# Myonuclear content regulates cell size with similar scaling properties in mice and humans

Kenth-Arne Hansson [1,2], Einar Eftestøl [1,3], Jo C. Bruusgaard[1,2,4], Inga Juvkam[1], Alyssa W. Cramer [3], Anders Malthe-Sørenssen[2,5], Douglas P. Millay [3,6] & Kristian Gundersen [1✉]

Muscle fibers are the largest cells in the body, and one of its few syncytia. Individual cell sizes are variable and adaptable, but what governs cell size has been unclear. We find that muscle fibers are DNA scarce compared to other cells, and that the nuclear number ($N$) adheres to the relationship $N = aV^b$ where $V$ is the cytoplasmic volume. $N$ invariably scales sublinearly to $V$ ($b < 1$), making larger cells even more DNA scarce. $N$ scales linearly to cell surface in adult humans, in adult and developing mice, and in mice with genetically reduced $N$, but in the latter the relationship eventually fails when they reach adulthood with extremely large myonuclear domains. Another exception is denervation-atrophy where nuclei are not eliminated. In conclusion, scaling exponents are remarkably similar across species, developmental stages and experimental conditions, suggesting an underlying scaling law where DNA-content functions as a limiter of muscle cell size.

[1] Department of Biosciences, University of Oslo, Oslo, Norway. [2] Center for Integrative Neuroplasticity, Department of Biosciences, University of Oslo, Oslo, Norway. [3] Division of Molecular Cardiovascular Biology, Cincinnati Children's Hospital Medical Center, Cincinnati, USA. [4] Department of Health Sciences, Kristiania University College, Oslo, Norway. [5] Department of Physics, University of Oslo, Oslo, Norway. [6] Department of Pediatrics, University of Cincinnati College of Medicine, Cincinnati, USA. ✉email: kristian.gundersen@ibv.uio.no

Why do organisms, cells and organelles have the absolute size they have, and how is size regulated? Throughout biology size matters, and the biologist and mathematician D'Arcy Wentworth Thompson succinctly expressed that perhaps the most immense challenge in science would be to unravel the mechanisms of size-regulation in biological systems[1].

Cellular function is tightly linked to the abundance of organelles, which typically grow in number or size to accommodate for the greater functional needs as cellular size increases. The change in number of organelles with a typical size parameter is called the scaling of that characteristic. Scaling laws typically reflect overall attributes of cells, and the cell's molecular or biochemical status represents the underlying manifestation of the observed type of scaling[2–8]. The scaling can express relationships of correlates to cell size by reporting the scaling exponent $b$ in a power function of the form $y = ax^b$.

The muscle fibers are the largest cells in the body. For example, the human sartorius muscle has an average fiber length of 42 cm[9] and a fiber cross-sectional area of about 2500 μm$^2$[10]. This leads to a total volume of 1050 nL, more than 4000 times that of the human egg cell. The muscle cells are also special in being one of just a handful of syncytia in the mammalian body. A human muscle cell might have more than 100 myonuclei per mm length of fiber[11,12], thus a sartorius cell might have more than 40,000 nuclei. We here investigate the scaling behavior of muscle cells in relation to their number of nuclei.

Little is known about the scaling principles related to syncytial cells. The high number of nuclei is believed to be necessary due to the vast cytoplasmic volumes and long transport distances. Thus, both a sufficient number of nuclei and optimal positioning of the nuclei are important to overcome these challenges[13–17], and hypertrophic growth are dependent on new nuclei from satellite cells[18–21].

A cytoplasmic-to-nucleus domain theory, postulating that each nucleus serves a certain cytoplasmic domain, has existed at least since the late nineteenth century[22]. We now know that each nucleus is surrounded by a synthetic machinery appearing to remain localized[23], and that many proteins are localized in the vicinity of the site of transcription[23–30].

As a consequence of the domain hypothesis it is often assumed that the nuclear number scale in direct proportion to cell volume, even if the absolute number of nuclei for fibers of the same size may vary between different muscle fiber types, age and previous history such as after strength-exercise where a "muscle memory" has been related to a persistently elevated number of myonuclei[13,15,20,31].

We here present data derived from 3D-reconstructions of fiber segments after confocal imaging of normal human and mouse fibers. While scaling studies so far has primarily been observational, we have also included material from mice with a conditional knock out reducing the number of cell nuclei in order to reveal mechanisms underlying size regulation.

We show that in all cases the relationship between the nuclear number and cytoplasmic volume adhered well to a relationship of the form:

$$N = aV^b,$$

where $N$ is the number of nuclei, $V$ the cytoplasmic volume, $a$ is a normalizing constant, and $b$ the scaling exponent. While this is a purely mathematical relationship which can be log transformed to yield a linear relationship, one can assign biological meaning to the constants[32–34].

The scaling exponent $b$ (the slope in a logarithmic plot) measure the magnitude of the increase in $N$ when $V$ increases. If $b = 1$ the number of nuclei increases in direct proportion to cell volume, and the myonuclear domains remain constant. In the present material $b < 1$ was observed for all experimental groups, indicating that when fiber volume increases the increase in the number of nuclei did not keep pace, and are thus insufficient to maintain constant myonuclear domain volumes. We hypothesize that the sublinear scaling of the number of myonuclei to cytoplasmic volume limits cell size. Variation in $a$ (the intercept of the y-axis in a logarithmic plot), while keeping $b$ fixed, reflects the differences in nuclear number for a given cell volume and thus the amount of cytoplasm each nucleus has to produce in order to achieve a muscle fiber of equal size. Thus, $a$ is inversely proportional to the nuclear cytoplasmic domain and could be considered as a "nuclear setpoint" for each nucleus ability to produce cytoplasmic volume.

## Results

**Number of nuclei scales sub-linearly to cell volume in mouse muscle fibers labeled in vivo.** To examine the relationship between nuclear number and size in vivo, we injected fluorescent oligonucleotides into 96 muscle cells (10–22 fibers per animal) from six female mice (age P70–77) in vivo and examined them in situ after fixation (see "Methods" section). This method enabled us to extract nuclear number and size-related parameters delineated by a single fiber membrane in continuous fiber segments (0.3–0.8 mm) and thus prevented labeling of e.g., satellite cells[13]. Segments encompassing the end-plate with the synaptic nuclei enclosed were excluded from further analysis. Although the synaptic nuclei represent only about 1% of the nuclei in a whole EDL fiber, in a smaller fiber segment this cluster of nuclei would constitute a larger proportion and might introduce variability in the counts.

By 3D confocal imaging, we visualized nuclei that with their characteristic shape were sharply delineated with intense labeling, while cell geometry was determined based on the fainter background staining of the cytosol (Fig. 1a). This allowed 3D reconstruction of the cell shape and the number and positions of the cell nuclei (Fig. 1b–d).

The injected fibers from each of the six muscles had an average cross-sectional area (CSA) of 1098 μm$^2$ ranging 866–1289 μm$^2$ (Fig. 1e). As a population of individual fibers, the CSA ranged 600–1600 μm$^2$ with a mean of 1109 μm$^2$ (95% CI, 1069–1149) (Fig. 1f). Similarly, the muscles had an average of 50 nuclei/mm (Fig. 1g) and displayed the same average when counted as single fibers (95% CI, 49, 52) (Fig. 1h). The myonuclear domain for the muscles averaged 22.3 pL (Fig. 1i), and when all the fibers were pooled (Fig. 1j), they displayed a right-skewed distribution with a mean of 22.4 pL (95% CI, 21.4, 23.2).

Myonuclear domains could also be expressed as surface domains which signify the surface area per nucleus, which were on average 2603 μm$^2$ (95% CI, 2396-2810) (Fig. 1k) and normally distributed (Fig. 1i)

In Fig. 1m–r individual fiber data are plotted as a function of different size parameters. The dashed blue lines represent the expected relationship if there were a strict linearity between number of nuclei and cell volume. It is evident that this relationship was not linear, and that the individual myonuclear domains increased with increasing size (Fig. 1n). When plotted and analyzed logarithmically (Fig. 1o), the relationship between number of nuclei and volume was nonlinear with $b = 0.73$ (95% CI, 0.65, 0.80) and statistically different from $b = 1$.

The surface area and nuclear number scaled linearly (Fig. 1p–q), and this agrees with analysis based on wide field microscopy[13], but the functional interpretation is not clear.

**Scaling behavior of adult human muscle are similar to mice in spite of differences in absolute numbers.** Next, we investigated the scaling behavior of fibers isolated from the m. vastus lateralis

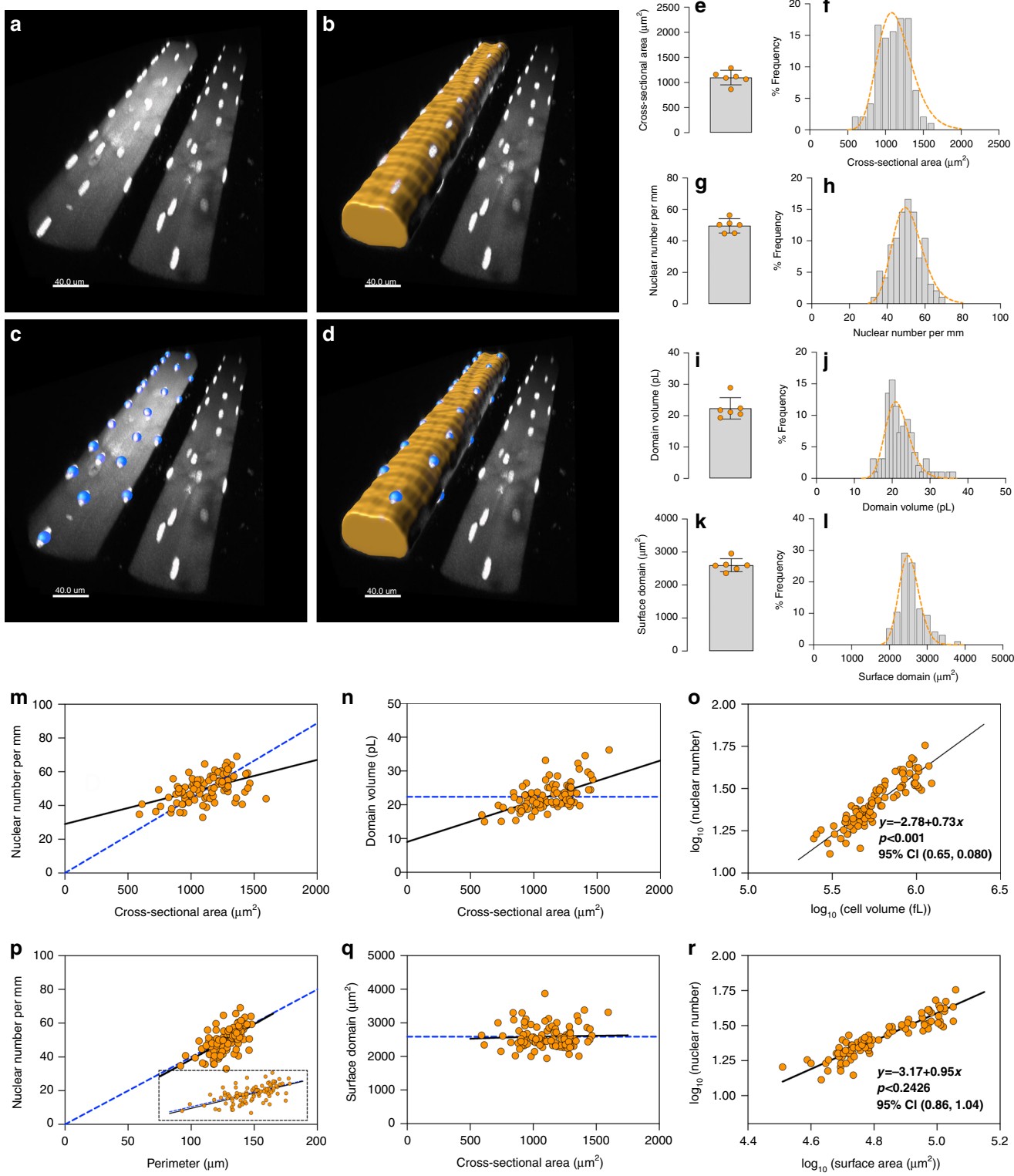

in human males, 20–29 years of age. Nuclei were labeled with DAPI, and autofluorescence was used to outline the fiber space allowing 3D reconstruction similarly to what was described for the mouse fibers above (Fig. 2a–d). The fibers were mounted on slides and the shape was frequently flattened compared to the in vivo observations.

The human fibers had CSAs more than twice as large as those from mice (Fig. 2e) and the individual fibers showed a bimodal distribution (Fig. 2f), with a mean of 2453 μm². The bimodal distribution was caused by two outlying individuals with lower values, and was not due to a bimodal distribution in each individual. The nuclear number per mm was three times as large as those found in mice (Fig. 2g, h), and individual fibers displayed a mean of 154 nuclei/mm (95% CI, 148–161). As a consequence, even if the fiber CSA was much larger in humans, the myonuclear domains were 29% smaller than in mice (Fig. 2i), and the individual fibers displayed sizes of domain volumes (Fig. 2j) that were fairly normally distributed around a mean of 16 pL (95% CI,

**Fig. 1 Nuclear number scales sublinearly to cell volume and linearly to fiber surface in mice labeled in vivo. a–d** Representative image from cells ($n = 96$) from the EDL muscle injected with a fluorescent dye that is taken up by the nuclei (**a**). **b** Background fluorescence was used to visualize the fiber volumes (orange). To calculate the average cross-sectional area and fiber volume the perimeter was outlined manually at 20–50 μm intervals. **c** Nuclear number was quantified by assigning a spot (blue) to each nucleus manually. **d** Rendered fiber (**b**) merged with the spots (**c**). **e, g, i,** and **k** mean value per muscle ($n = 6$), **f, h, j,** and **l** show the frequency distribution per fiber ($n = 96$), for cross-sectional area (**e, f**), nuclear number per mm (**g, h**), domain volumes (**i, j**), and surface domains (**k, l**). **m** Nuclear number per mm versus cross-sectional area were statistically tested against linear scaling ($b = 1$, dashed blue line). Comparison of fits gave a F-value of 53.74 ($p < 0.0001$). **n** Nuclear domains versus cross-sectional area were statistically tested against the dashed line which indicate a fixed scaling ($b = 0$). Comparison of fits gave a F-value of 65.80 ($p < 0.0001$). **o** Nuclear number versus cell volume plotted and analyzed in log–log space gave a slope of $b = 0.73$ (95% CI: 0.65, 0.80). **p** Nuclear number per mm versus the fiber perimeter were statistically tested against a linear relationship (dashed blue line). Comparison of fits yielded a F-value of 0.1249 ($p = 0.7245$). **q** Surface domains versus cross-sectional area tested against a horizontal slope ($b = 0$, dashed blue line), gave an F- value of 0.3070 ($p = 0.5808$). **r** Nuclear number versus surface area plotted in log–log space gave a slope of $b = 0.95$ (95% CI: 0.86, 1.04). Error bars in **e, g, i,** and **k** represent the 95% CI's, while the nonlinear lines (orange) in **f, h, j,** and **l** were fitted by a Gaussian function. In **m–r** regression lines were fitted with an OLS method with (1, 94) degrees of freedom and compared with extra sum-of-squares F-test. Scale bars, 40 μm in **a–d**. Source data are provided as a Source Data file.

15.6–16.5). Also, the surface domains were smaller than in mice with an average 1565 μm² (95% CI, 1335–1795) (Fig. 2k, l).

As in mice, the number of nuclei increased sublinearly with volume (Fig. 1m–o), thus in a logarithmic plot (Fig. 2o) the exponent was $b = 0.66$ (95% CI 0.60–0.72). The human fibers displayed smaller myonuclear domain volumes, relative to mice, for a given cross-sectional area (compare Fig. 1j and Fig. 2j). As in mice the number of nuclei scaled roughly in direct proportion to fiber surface (Fig. 2p–r).

**Myonuclei are not lost by denervation, but the sublinear scaling behavior is exaggerated by differential atrophy.** We next studied the scaling behavior in adult mouse muscles after 14 days of denervation. The CSA was reduced from a mean of 1098 μm² (95% CI 953, 1243) to 567 μm² (95% CI 473–660) (Fig. 3a). The number of nuclei per mm was not affected by the denervation (Fig. 3b, f), in agreement with our previous conclusions from wide field microscopy[35–38], but as this is the first time confocal imaging is applied to in situ preparations of denervated muscle, it strengthens the notion that myonuclei are not lost during denervation atrophy.

The distribution of nuclear number remained unaltered (Fig. 3f), while the distribution of CSA (Fig. 3e) and nuclear domains (Fig. 3g, h) shifted left but remained close to a normal distribution. When we analyzed nuclear number versus fiber volume, denervation changed the scaling exponent from $b = 0.73$ in the normal fibers to $b = 0.36$ (95% CI 0.28–0.44) (Fig. 3k) in fibers of denervated muscles.

Since the number of nuclei was unaffected by denervation, this suggests that the fibers with smaller volumes at the time of denervation atrophied proportionally more than the larger fibers. Please note that on average shorter fiber segments were labeled in the denervated fibers due to slower diffusion of the dye in the thin fibers, hence the absolute number of nuclei were lower (Fig. 3k). This does not indicate a loss of nuclei as is evident from Fig. 3b, f. As opposed to the scaling of the other groups in the present paper, number of nuclei did not scale linearly to fiber surface area ($b = 0.64$) after denervation (Fig. 3n). Since the number of nuclei is not altered by denervation (Fig. 3b), this variable reflected the surface area the fiber had before the denervation, rather than the current area. It thus shows that the scaling properties are not retained during denervation atrophy.

**Scaling behavior is retained during postnatal growth, also after genetically inhibiting myonuclear accretion.** We finally investigated developing mouse muscles with and without preventing satellite cell fusion by genetic deletion of myomaker, a membrane protein required for fusion of satellite cells[38,39]. The deletion was

initiated at postnatal day P6; in a transgenic mouse line named Δ2w in the accompanying paper[40].

We observed no effect of the deletion on fiber size when observed at P13, in both groups the fibers were very small with a CSA average of 236 μm² (95% CI 224–247) for the controls and 225 μm² (95% CI 211–238) for the Δ2w mice (Fig. 4a). Both the controls and Δ2w fibers grew from P13–P27 and the size distribution remained similar (Fig. 4a, b). After P35 there was less radial growth in both groups, but at P35 the control fibers had much larger CSAs (764 μm², 95%, 729–799) than the Δ2w fibers (540 μm², 95% CI 555, 625) (Fig. 4a, b). At all time-points the number of nuclei per mm remained lower in the Δ2w mice compared to controls, and the difference became larger at P35 corresponding with the control fibers displaying an increase in nuclear density (Fig. 4c). In adults at P150 the nuclear number was 127% higher in the controls compared to the Δ2w mice. The increase in the radial size during maturation was paralleled by larger nuclear domains in both groups during growth, but even more so in the Δ2w group which at P150 had about 70% larger domain volumes than the controls (Fig. 4d). The development in surface domains was similar and at P150 the Δ2w surface domain areas was 95% larger than in control fibers (Fig. 4e).

For number of nuclei and cell volume the scaling behavior was similar in controls and the Δ2w mice for ages P13–P42. At all ages $b < 1$, and at P13–P42 the range was 0.52–0.72. During development, constant $a$ reflecting a difference in nuclear number and the absolute ability of the nuclei to "produce" cytoplasm became increasingly more different between Δ2w and control mice. Thus, Δ2w cells needed to produce more cytoplasm per nucleus for any given cross-sectional area (Fig. 4d), but the slope ($b$) of the logarithmic curves for the two groups was similar at each timepoint, i.e., the regression lines were close to parallel in spite of shifts in absolute values (Fig. 5a). A shallower slope at P150 compared to P13 suggest that the growth rate is larger for larger cells compared to smaller cells during this period. In adults (P150) the controls had a similar slope as found earlier in development ($b = 0.62$), but in Δ2w mice it decreased dramatically ($b = 0.44$), similar to the denervated fibers ($b = 0.36$), where it reflected a disproportionate atrophy of smaller fibers relative to larger fibers. In the Δ2w mice at P150, however, the shallow slope might reflect that larger fibers grew at a greater rate between P42 and P150 compared to large control fibers. This difference in scaling properties between nuclear number and fiber volume at P150 indicates that the discrepancy in domain volumes between controls and Δ2w mice were much greater in larger fibers relative to smaller ones.

When analyzing the relationship between nuclear number and surface area (Fig. 5b), a similar picture emerged, and notably for all ages, in both controls and Δ2w fibers, the scaling between

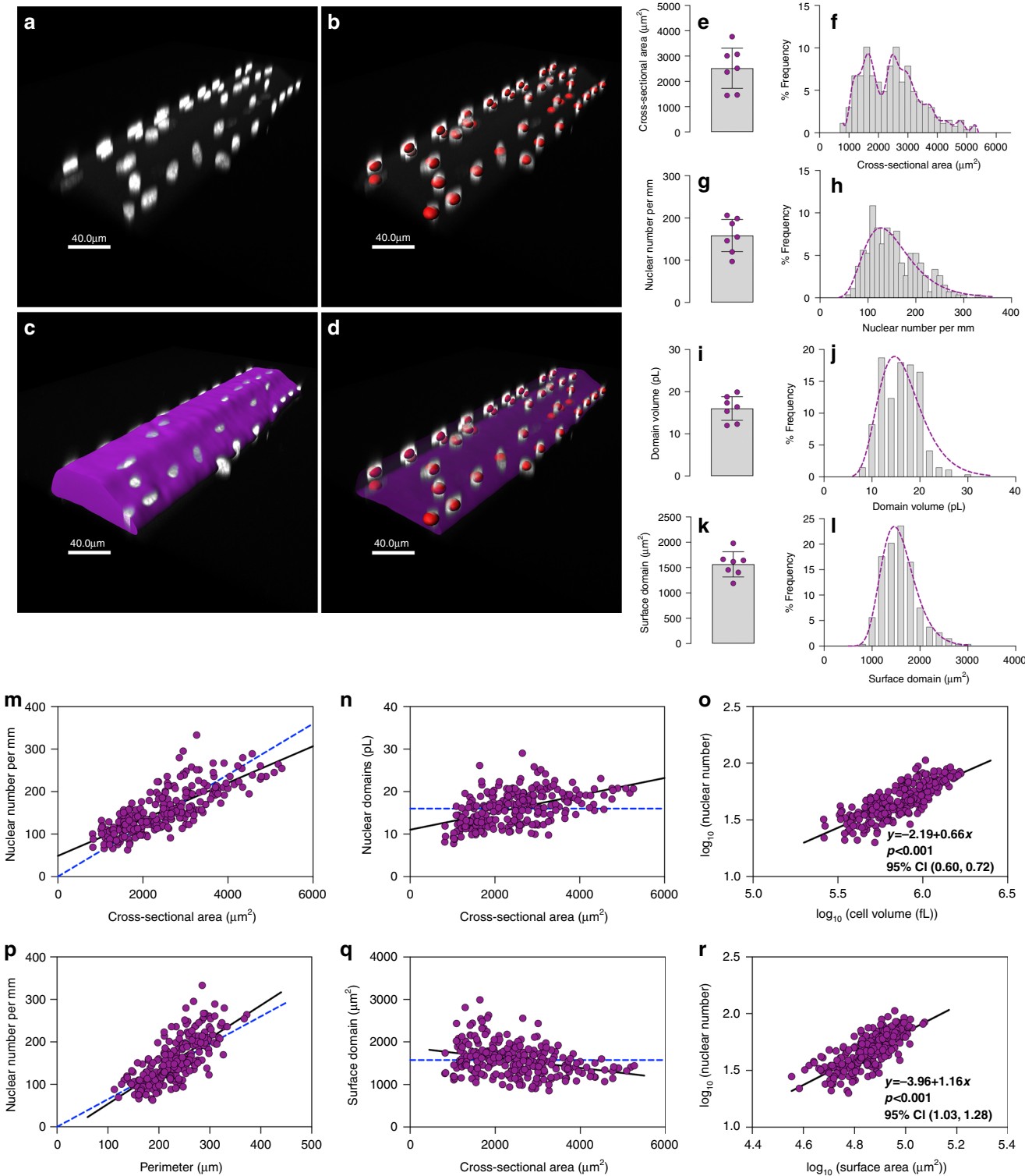

nuclear number and cell surface did not deviate significantly from a linear relationship ($b \approx 1$).

One might speculate why the similarity in scaling properties between nuclear number and fiber volume breaks down after P42, and it is noteworthy that the nuclear domains became very large in the adult Δ2w mice at P150 (Fig. 4d), the domain volumes were 70% larger and the surface domain areas 95% larger than in the controls, and this might somehow perturb the normal scaling relationship.

One mechanism that might influence scaling of nuclear number and cell size would be if the quality of each nucleus was altered e.g., by increasing the DNA content by endoreplication or unpacking of chromatin. Thus, it was recently shown that nuclear size was positively correlated with fiber size in muscles of the Drosophila larvae[41]. In order to investigate if similar mechanisms were operating in our material, we 3-D rendered the nuclei (Fig. 6a) and compared the DNA volumes as labeled by DAPI. These measurements were performed in controls and Δ2w

**Fig. 2 Scaling properties of adult human muscle are similar to mice in spite of differences in absolute numbers. a–d** Representative image from muscle cells ($n = 267$) from the human v. lateralis labeled with DAPI to visualize nuclei (**a**). **b** Nuclear number was quantified by assigning a spot (red) to each nucleus. **c** Background fluorescence was used to automatically 3-D render the cells´ morphology. **d** Shows a 3-D rendered transparent muscle cell with its nuclei. **e, g, i,** and **k** Highlights the mean (arithmetic) value per muscle ($n = 7$), while **f, h, j** and **l** show the frequency distribution per fiber ($n = 267$), for cross-sectional area (**e, f**), nuclear number per mm (**g, h**), domain volumes (**i, j**) and surface domains (**k, l**). **m** Nuclear number per mm versus cross-sectional area tested against linear scaling ($b = 1$, dashed blue line). Comparison of fits gave a $F$-value of 82.39 ($p < 0.0001$). **n** Nuclear domains versus cross-sectional area were tested against the dashed line which indicate a fixed scaling ($b = 0$). Comparison of fits gave the $F$-value of 101.1 ($p < 0.0001$). **o** Nuclear number versus cell volume plotted and analyzed in log–log space gave a slope of $b = 0.66$ (95% CI: 0.60, 0.72). **p** Nuclear number per mm versus the fiber perimeter were statistically tested against a linear relationship (dashed blue line). Comparison of fits yielded a $F$-value of 5.967 ($p = 0.0152$). **q** Surface domains versus cross-sectional area tested against a horizontal slope ($b = 0$, dashed blue line), gave an $F$-value of 29.55 ($p < 0.0001$). **r** Nuclear number versus surface area plotted in log–log space gave a slope of $b = 1.16$ (95% CI: 1.03, 1.28). Error bars in **e, g, i,** and **k** represent the 95% CI's, while the non-linear lines (purple) in **h, j,** and **l** were fitted by a Gaussian function, and **f** by a locally weighted smoothing regression (LOWESS). In **m–r** regression lines were fitted with an OLS method with (1, 265) degrees of freedom and compared with extra sum-of-squares $F$-test. Scale bars, 40 μm in **a–d**. Source data are provided as a Source Data file.

mice at P150, which had the largest disparity in myonuclear domains, and at P27 which displayed the fastest growth.

The nuclear volumes were very similar at the two timepoints and no difference between controls and Δ2w mice was discernible (Fig. 6b–d). Moreover, there was no correlation between nuclear volume and fiber domain volume in any of the groups (Fig. 6e, f).

## Discussion

By precise measurements of 3D-reconstructed fibers we show that the scaling properties are shared by muscle fibers from normal adult mice and humans, as well as by growing mouse fibers during postnatal development, with and without a genetically reduced number of myonuclei. In all cases the scaling exponent between nuclear number and cell volume was significantly lower ($b < 1$) than a linear relationship (Fig. 5a). Thus, larger cells had larger myonuclear domain volumes, and the ever increase in domain volumes when going from small to larger fibers might be a general limiter of fiber growth.

The scaling exponents for nuclear number and cell volume were similar for the different experimental groups with an average of $b = 0.64$ (Fig. 7) (excluding denervated muscles and Δ2w muscles at P150 since they had significantly lower exponents, discussed below), this is close to $b = 0.67$ (Fig. 7), which would be the exponent if the nuclear number scaled in direct proportion to the cell surface, and in fact for all the experimental groups (except denervation, discussed below) the scaling relationship between number of nuclei and cell surface was not significantly different from a linear relationship.

Based on these findings, we conclude that at various timepoints during development fiber surface grows linearly to the number of nuclei, but with different absolute sizes of surface domains (i.e., differences in the a-setpoint). Fibers with various nuclear number (small versus large and Δ2w versus normal) develops over time to reach individual size-plateaus and those fibers with a higher biosynthetic capacity (i.e., nuclear number) grows faster. However, the muscles must be able to adapt and respond over time to satisfy new environmental conditions[42–44]. For example, muscle fibers maintain an ability to alter size as a response to stimuli such as strength exercise[16]. Such changes are likely to involve signals crossing the cell surface, which might regulate the accretion of cell nuclei which again regulate and limit cell size (Fig. 8). As opposed to what was observed for developmental growth, and for the adaption to a reduced number of nuclei in the Δ2w mice, such processes do not involve significant changes in each nucleus ability to produce volume (i.e., changes in the setpoint a).

The exceptions to the similarities in scaling exponents, were denervated fibers ($b = 0.36$), and fibers from the Δ2w mice at P150 ($b = 0.44$). In these cases, there was a much shallower relationship between nuclear number and size. Under both these experimental

conditions the nuclear number was relatively constant during the period of size change (i.e., atrophy and developmental growth, respectively). The shallower relationship at the later developing stages of the Δ2w mice might reflect that larger cells grew more compared to smaller cells between P42 and P150. For denervated muscles, larger cells seemed to atrophy less than smaller cells. After denervation the number of nuclei also does not reflect the current size, since nuclei are not lost during atrophy[35], but is rather a cytoarchitectural reminiscence of the cell size before the nerve transection[16]. It was not feasible to use the microinjection technique to label the very thin fibers after long-term denervation, but it has previously been observed in isolated fiber segments that the nuclei are preserved for as long as 4 months of denervation[45].

The similarities in scaling behavior across our diverse experimental groups is striking given that they vary in species, sex, age, muscle, and fiber type composition. While we did not directly measure fiber type in our studies, our injections were all in fibers at the lateral surface of the EDL they are virtually all of type 2b, while the developing EDL material is a mixture of type 2b and 2x (see "Methods" section). The humans were all adult males, and the vastus lateralis biopsies contained an equal mixture of type 1 and 2 (see "Methods" and ref. [12]).

The variability in these attributes lead to differences in absolute values for nuclear number, cell sizes, and myonuclear domains, but the scaling behavior is preserved supporting the idea of a fundamental biological relationship between DNA-content and cell size as suggested by Cavalier–Smith and Gregory[46–48]. For example, Gregory found a clear positive correlation between cell sizes and DNA content of diploid cells from 159 species of vertebrates. In humans a similar correlation was found between DNA content and cell size, across haploid cells, polyploid cells, and polykaryons with a scaling exponent of $b = 0.74$[49]. Muscle fibers were not included in their analysis, but our data suggests that they are adhering to similar scaling properties.

Interestingly, if we down-scale the relationship of the human muscle cells investigated in the present study, based on the proportionality $N \propto V^{0.67}$, to a theoretical muscle cell having only a single diploid nucleus, this cell would exhibit a cell volume of ~2.2 pL which is comparable to human lymphocytes and monocytes[49–52]. If, however, making a theoretical up-scaling based on the scaling exponent, starting with a typical diploid human cell containing 7 pg of DNA, to sizes equivalent to average muscle cell volumes, the DNA content carried would be equal to 2700 pg/mm fiber or about 390 diploid human nuclei. This is about 2.5 times the nuclear number we observed. The average nuclear domain volume we observed in human muscle is comparable to the volume of polyploid megakaryocytes, but they are carrying a DNA content of about 45 pg, more than six times that of a single diploid nucleus[53].

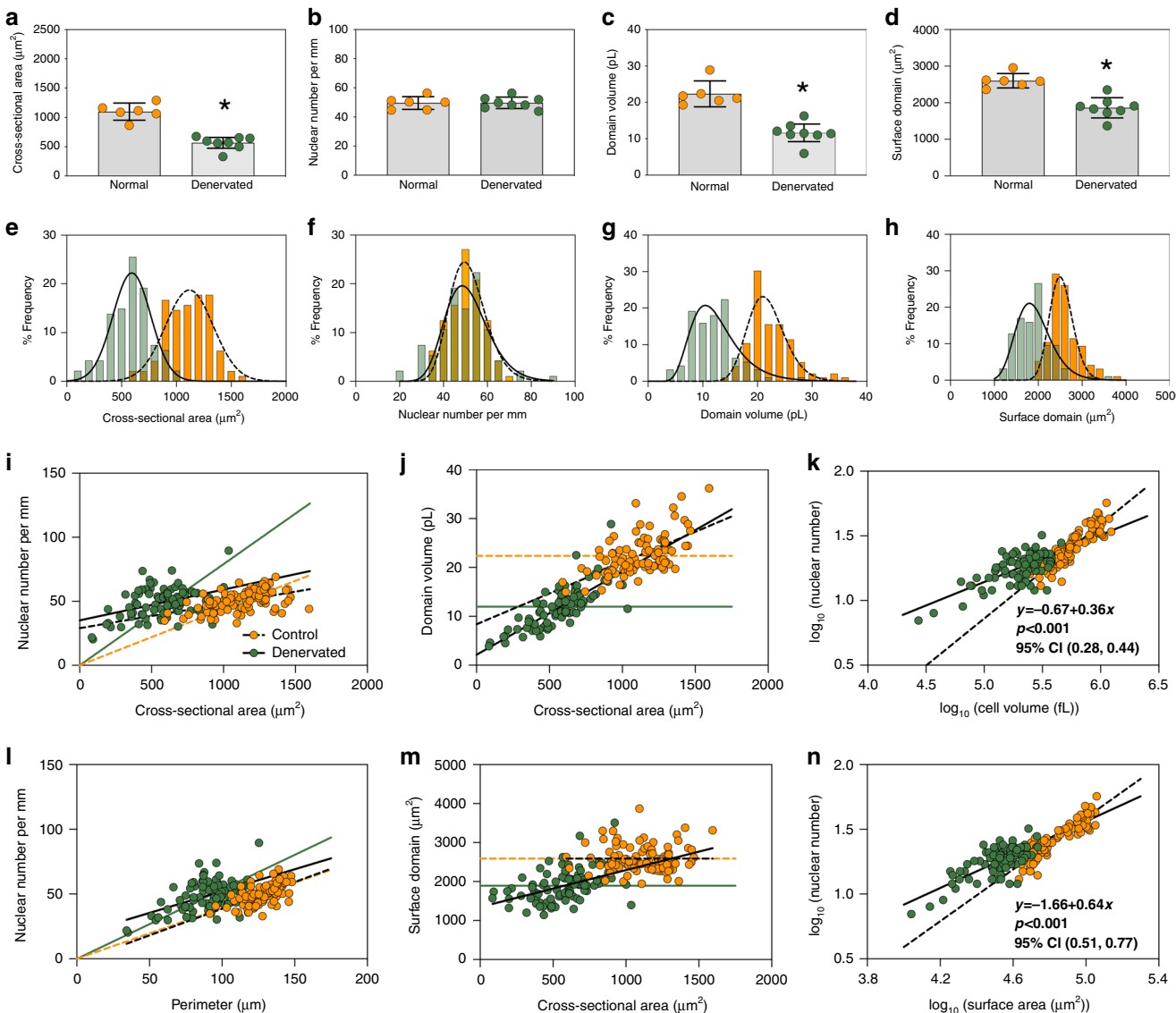

**Fig. 3 Relationship between nuclear number and cell size after 14 days of denervation. a–d** Highlights the mean (arithmetic) value per muscle, while **e–h** shows the frequency distribution per fiber, for cross-sectional area (**a**, **e**), nuclear number per mm (**b**, **f**), domain volumes (**c**, **g**), and surface domains (**d**, **h**) in denervated muscle (green) and normal muscle (orange, these are the same data as presented in Fig. 1). Asterisks (*) indicate significantly different means based on results from two-sided Welch's $t$-test: **a** ($p < 0.0001$, $t = 7.707$, and $df = 9$), **b** ($p = 0.9719$, $t = 0.03611$, and $df = 10$), **c** ($p = 0.0001$, $t = 6.220$, and $df = 9$) and **d** ($p < 0.0001$, $t = 5.857$, and $df = 12$). **i–n** Relationships of nuclear number and fiber size analyzed for denervated muscle (green) compared to normal muscles (orange). Statistics for the normal muscles are presented in Fig. 1. **i** Nuclear number per mm versus cross-sectional area for denervated fibers (solid black line) tested against a linear scaling ($b = 1$, green solid line). Comparison of fits gave a $F$-value of 109.9 ($p < 0.0001$). **j** Domain volumes versus cross-sectional area were tested against the solid green line ($b = 0$). Comparison of fits gave an $F$-value of 153.1 ($p < 0.0001$). **k** Nuclear number versus cell volume plotted and analyzed in log–log space gave a slope of $b = 0.36$ (95% CI: 0.28, 0.44). **l** Nuclear number per mm versus the fiber perimeter were statistically tested against a linear relationship (solid green line). Comparison of fits yielded a $F$-value of 11.51 ($p = 0.0010$). **m** Surface domains versus cross-sectional area tested against a horizontal slope ($b = 0$, solid green line), gave an $F$-value of 23.85 ($p < 0.0001$). **n** Nuclear number versus surface area plotted in log–log space gave a slope of $b = 0.64$ (95% CI: 0.51, 0.77). In **i–n** regression lines were fitted with an OLS method with (1, 92) degrees of freedom and compared with the extra sum-of-squares $F$-test. $n = 8$ (denervated) and 6 (normal) muscles for **a–d**, while $n = 94$ (denervated) and 96 (normal) cells for **e–h**. Error bars in **a–d** represent the 95% CI and the curves in **e–h** were fitted by a Gaussian function. Source data are provided as a Source Data file.

These comparisons demonstrate that muscle fibers in general, and in particular large fibers, have a very low DNA content per volume compared to other cell types, and supports the idea that DNA (i.e., number of nuclei) could be a limiting factor in upscaling muscle cells.

On the other hand, the present data does not support a simple notion of an absolute myonuclear domain size. For example, the logarithmic plots for developing muscles with and without impaired myonuclear fusion suggest that the scaling relationship was similar, but the $\Delta 2w$ fibers had larger myonuclear domains, and seemed to be regulated with a higher absolute cytoplasmic "setpoint" (reflected by a change in the constant "a") assigned to each nucleus (Fig. 5b). While this demonstrates a flexibility in nuclear domains, it should be noted that the reduced number of nuclei led to the $\Delta 2w$ having smaller radial sizes compared normal cells (Fig. 4b).

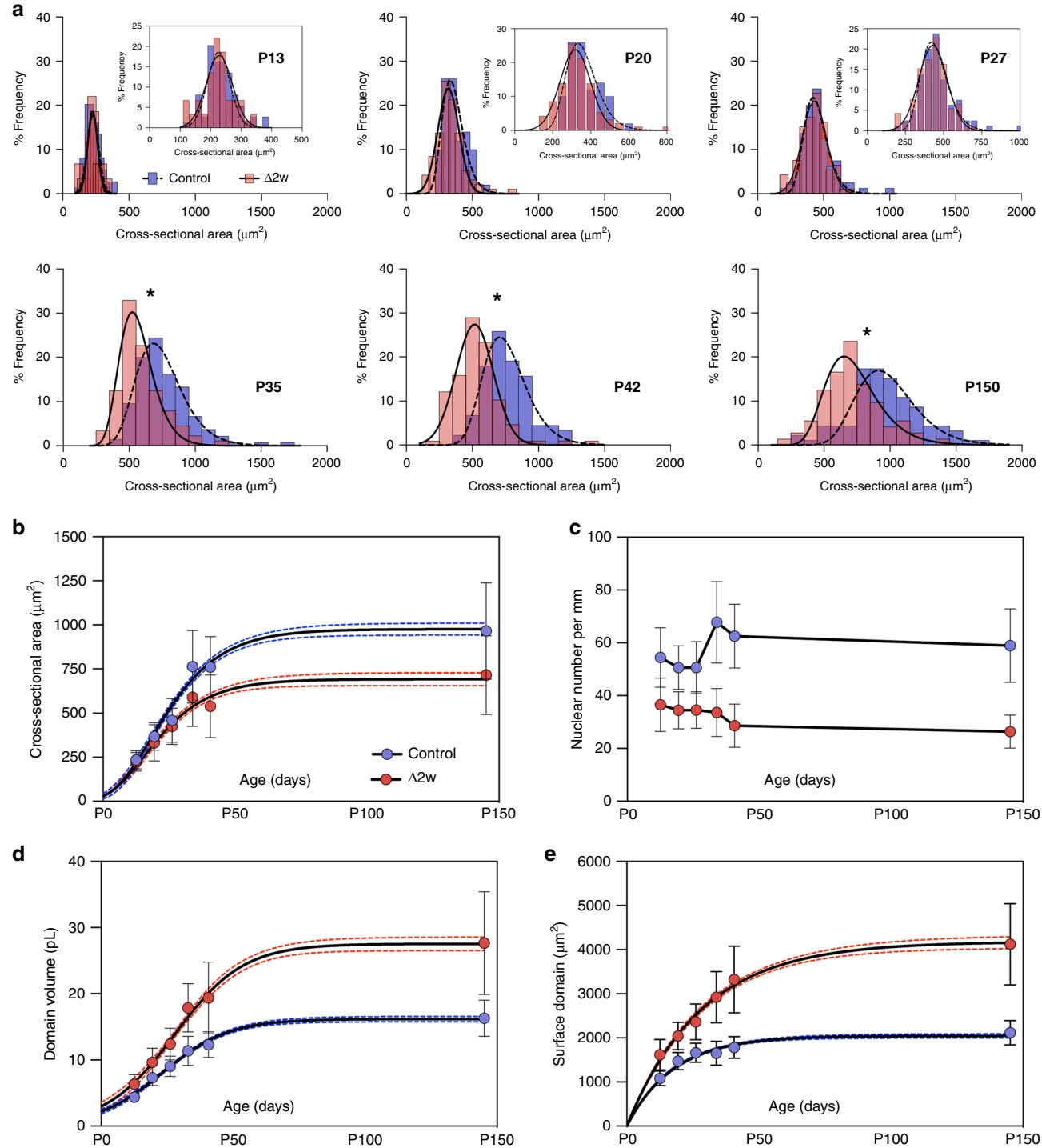

**Fig. 4 Inhibiting myonuclear accretion genetically causes smaller muscle fibers and larger myonuclear domains during development. a** Frequency distribution of cross-sectional area from P13 to P150 between normal EDL cells (blue, dashed Gaussian line) and Δ2w cells (red, solid Gaussian line). Inset in P13–P27 shows the zoomed distribution. Asterisks (*) indicate nested two-sided $t$-test results: P35 ($p = 0.0075$, $t = 3.56$, and df = 8), P42 ($p = 0.0067$, $t = 4.45$, and df = 5) and P150 ($p < 0.0001$, $t = 6.29$, and df = 162), while for the remaining age-matched comparisons: P13 ($p = 0.4838$, $t = 0.7708$, and df = 4), P20 ($p = 0.2830$, $t = 1.239$, and df = 4) and P27 ($p = 0.8157$, $t = 0.2488$, and df = 4). **b–e** Size-related parameters plotted against age for cross-sectional area (**b**), nuclear number (**c**), nuclear domains (**d**), and fiber surface domains (**e**). Solid lines in **b** and **d** were fitted to data using a logistic growth model, while **e** was fitted by an exponential plateau function. Dashed lines represent 95 % CI and each data point represents the mean value with the standard deviation as error bars. df = degrees of freedom. All data in **a**–**e** were calculated from the following number of control cells: P13 = 74 cells, P20 = 50 cells, P27 = 80 cells, P35 = 135 cells, P42 = 89 cells, and P150 = 92 cells. The number of myomaker Δ2w cells were; P13 = 59 cells, P20 = 94 cells, P27 = 92 cells, P35 = 88 cells, P42 = 107 cells, and P150 = 72 cells. Source data are provided as a Source Data file.

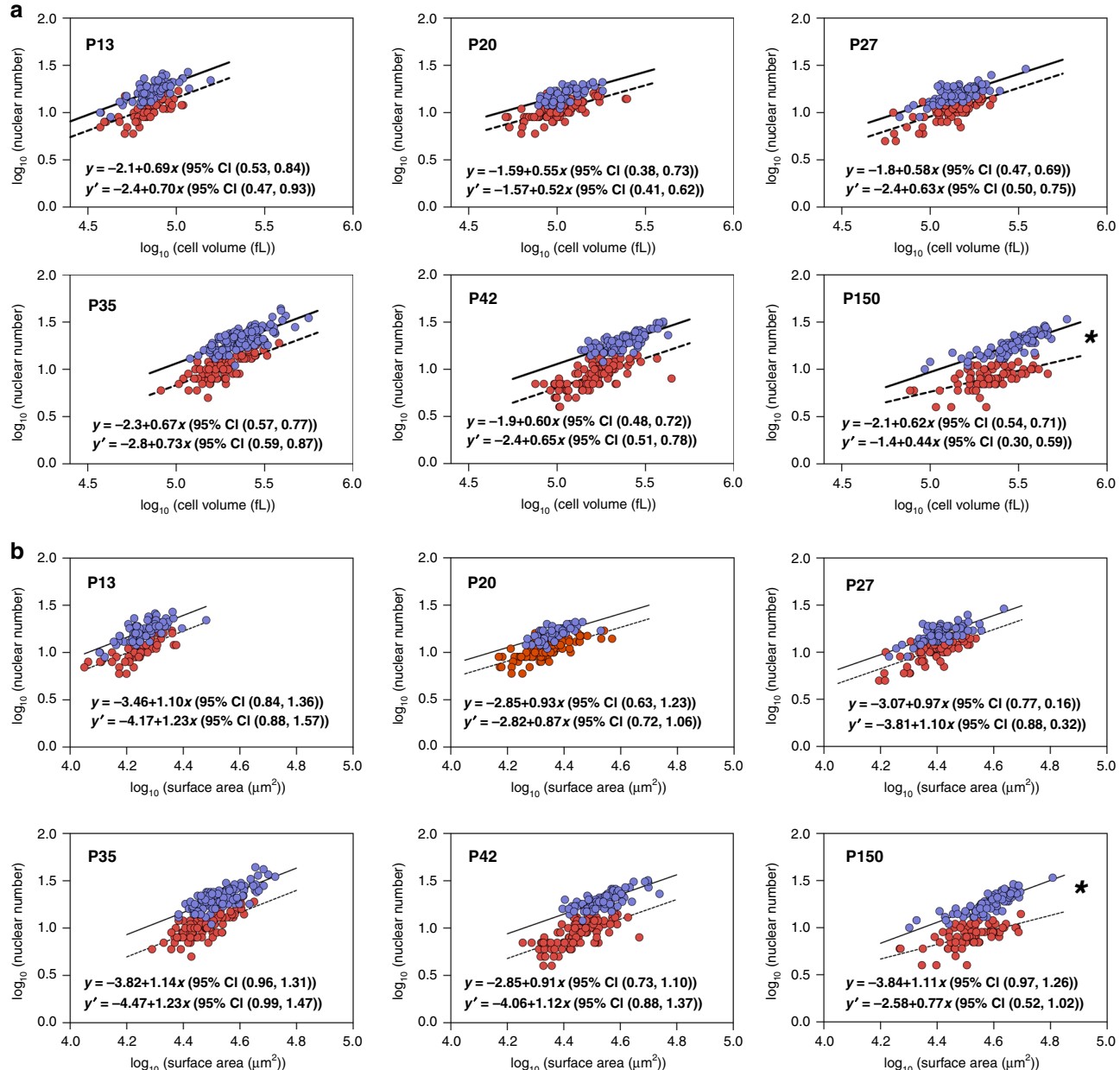

**Fig. 5 Scaling behavior is retained during post-natal growth, also after inhibiting myonuclear accretion genetically. a** Nuclear number versus cell volume in log–log space plotted from P13 to P150 for normal EDL cells (blue) and Δ2w cells (red). Solid lines (control) and dashed lines (Δ2w) were fitted after an OLS regression. Between group comparisons of the scaling exponent gave a *p*-value for each age as: P13 ($p = 0.9027$), P20 ($p = 0.7705$), P27 ($p = 0.6204$), P35 ($p = 0.4528$), P42 ($p = 0.6731$), and P150 ($p = 0.0295$, * statistically different at $p < 0.05$). The log–log model after fitting for normal cells is highlighted by **y**, while **y′** represent the equation for Δ2w cells. **b** Nuclear number versus cell volume in log–log space plotted from P13 to P150 for normal EDL cells (blue) and Δ2w cells (red). Solid lines (control) and dashed lines (Δ2w) were fitted after an OLS regression. Comparisons of the scaling exponent between the two groups gave a *p*-value for each age as: P13 ($p = 0.5471$), P20 ($p = 0.8265$), P27 ($p = 0.3758$), P35 ($p = 0.5246$), P42 ($p = 0.2006$) and P150 ($p = 0.0170$, *statistically different at $p < 0.05$). The log–log model after fitting for normal cells is highlighted by **y**, while **y′** represent the equation for Δ2w cells. In **a** and **b**, the number of control cells for each age were: P13 = 74 cells, P20 = 50 cells, P27 = 80 cells, P35 = 135 cells, P42 = 89 cells, and P150 = 92 cells. The number of myomaker Δ2w cells were; P13 = 59 cells, P20 = 94 cells, P27 = 92 cells, P35 = 88 cells, P42 = 107 cells and P150 = 72 cells. All *p*-values were extracted from the extra sum-of-squares *F*-test. Source data are provided as a Source Data file.

As discussed above all muscle fibers are remarkably low in DNA content compared to other cell types, but for the Δ2w mice the situation was even more striking, and one might have expected a functional "penalty" since excessively diluted cytoplasm (i.e., the number of organelles do not keep a commensurable pace with the cellular volume) leads to disruption of cellular processes[54–56]. Similarly, in most previous cases where muscle fibers were manipulated to have large myonuclear domains, function was impaired[14,57–59].

For example, Omairi et al.[57] showed that myostatin null mice in spite of having larger EDL muscles did not display higher maximal isometric tetanic force, and thus reduced specific force due to a disproportional growth of cell size compared to the content of myofibrils, and the fiber also had abnormal ultrastructure. When Estrogen-related receptor gamma (Errγ) was overexpressed in these mice, the number of myonuclei was increased and the specific force and other functional properties were restored.

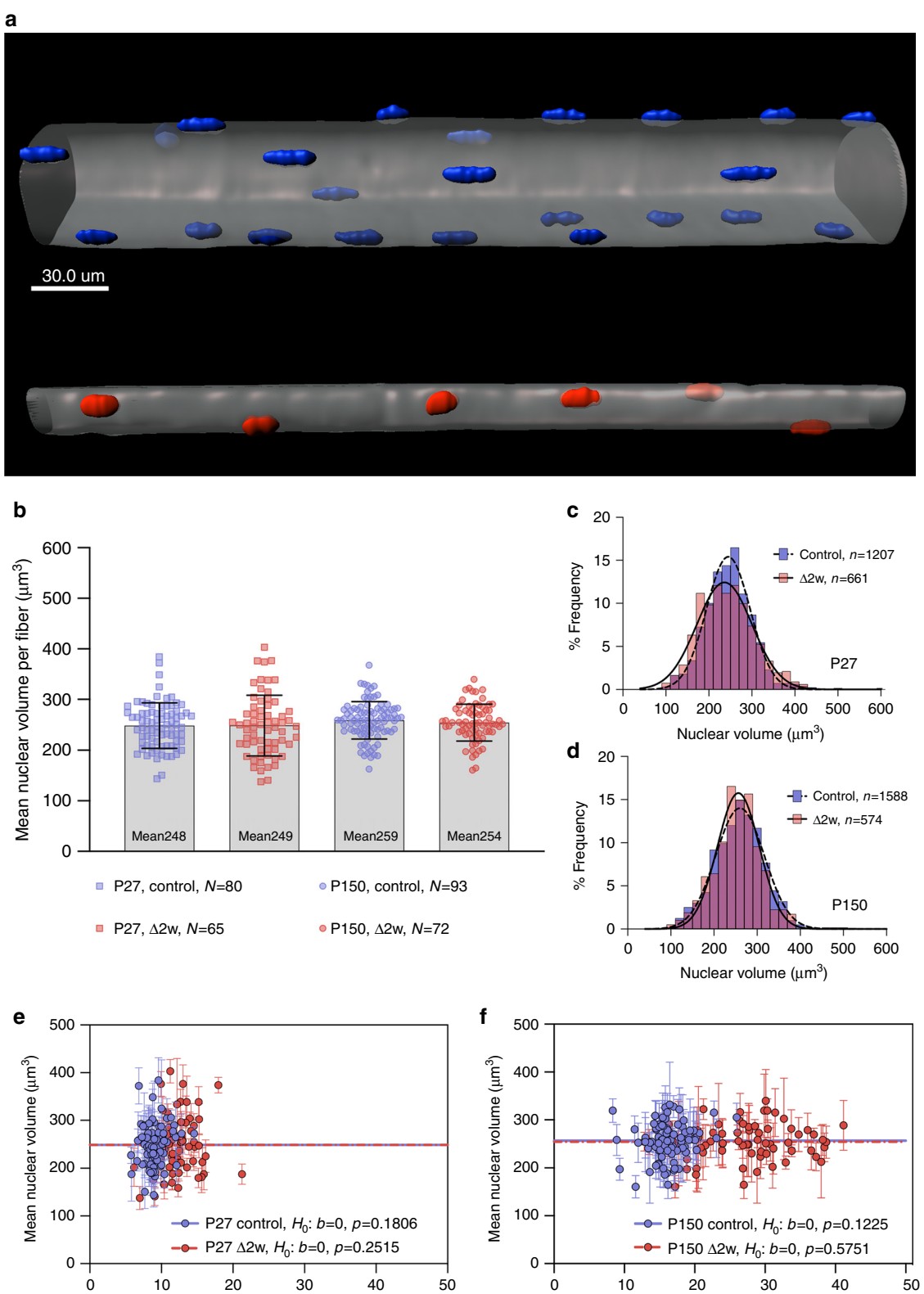

While the Δ2w muscles displayed normal specific force and fatigability[40], we cannot exclude the presence of some form of functional impairment for example if the muscles were put under stress such as increased metabolic or mechanical demand during exercise.

Nevertheless, in regulation of size in the strict sense the differences between the normal and Δ2w mice during development suggest that under these conditions, the flexibility in myonuclear domains were considerable. For example, at P150, if we downscaled the size of the control cells, to be as small as the Δ2w cells, they exhibit an average of 48 nuclei per mm, which were higher by a factor of 1.85 compared to the average nuclear number (26 nuclei per mm) in Δ2w cells. If upscaling their size to achieve domain volumes equivalent to values of Δ2w at P150

**Fig. 6 Nuclear volume is fixed across development and scale invariant with domain volumes. a** Nuclei and fiber 3D rendered to illustrate the difference in nuclear number and fiber size between control fibers (blue nuclei) and Δ2w fibers (red nuclei). Nuclear volume was determined based on fluorescence from DAPI stained nuclei. Nuclei that were positioned at fiber ends and only partially visualized were excluded from quantification. **b** Mean nuclear volume per fiber at P27 and P150 between control (blue) and Δ2w (red) were statistical non-significant when tested for differences among group means with a Brown–Forsythe (F-value of 0.93333 (DF = 3, df = 223), p = 0.4254) and Welch's ANOVA test (F-value of 1.049 (DF = 3, df = 159), p = 0.3756). **c** Frequency distribution of nuclear volume plotted independent of fiber at P27 and P150 (**d**). Mean nuclear volume versus domain volume at P27 (**e**) and P150 (**f**) were statistically tested against a horizontal slope (b = 0) which gave a F-value of 1.825 and 1.339 at P27 for controls and Δ2w fibers respectively. At P150 the F-value were 2.434 and 0.3173 for the Δ2w. All comparisons were statically non-significant with the corresponding p-value as highlighted in **e** and **f**. Source data are provided as a Source Data file.

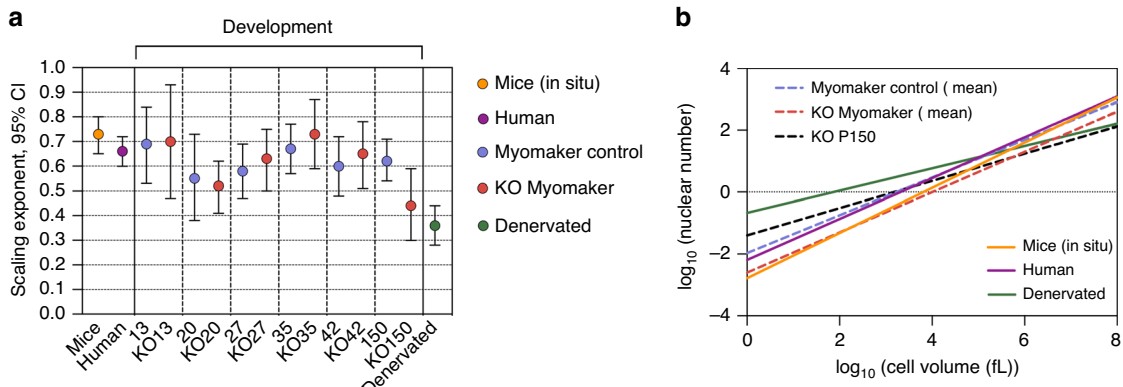

**Fig. 7 Scaling relationship of nuclear number and cell volume summarized. a** All scaling exponents extracted with their 95% CI. **b** Nuclear number versus cell volume plotted in log–log space for mice ($\log_{10}$ [nuclear number] = −2.78 + 0.73 $\log_{10}$ [cell volume]), humans ($\log_{10}$ [nuclear number] = −2.19 + 0.66 $\log_{10}$ [cell volume]), myomaker control cells ($\log_{10}$ [nuclear number] = −1.97 + 0.61 $\log_{10}$ [cell volume]), Δ2w cells without P150 ($\log_{10}$ [nuclear number] = −2.60 + 0.65 $\log_{10}$ [cell volume]), Δ2w P150 ($\log_{10}$ [nuclear number] = −1.4 + 0.44 $\log_{10}$ [cell volume]) and denervated cells ($\log_{10}$ [nuclear number] = −0.67 + 0.36 $\log_{10}$ [cell volume]).

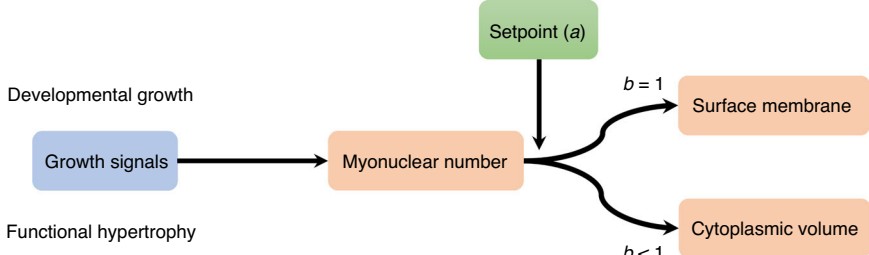

**Fig. 8 Model for regulation of muscle fibers size related to scaling relationship.** Our model proposes that the number of myonuclei in a fiber segment is determined by growth signals during development and hypertrophy. The number of nuclei and a setpoint a (which is a productivity number for each nucleus) determine cellular dimensions. The number of nuclei scales invariably sublinearly to cell volume (b < 1), but linearly to cell surface (b = 1).

(28 pL), controls would reach a cross-sectional area of astonishing 4700 μm² , which is larger by a factor of 4.9 when compared to the observed average cross-sectional area (966 μm²) and twice as large as the human fibers. In other words, nuclei in Δ2w cells have expanded their cytoplasmic domain to achieve larger fiber sizes, and thus demonstrate the ability to "produce" volume under these experimental conditions.

Commoner was the first to theorize that DNA content control cell size[60], but in spite of the many studies pointing towards an obvious positive relationship between the DNA content and cell size the causality is far from clear and it is still debated whether DNA content controls cell size, or cell size controls DNA content[49]. We will argue that the former mechanism is prevalent. Disuse atrophy does not lead to elimination of nuclei as seen from the current study (also see refs. [35,45,61,62]), during hypertrophy myonuclei precedes hypertrophy during hypertrophic growth[37], and hypertrophy is largely prevented when accretion of myonuclei is prevented[18–21]. In these experiments there were apparently

little or no flexibility in myonuclear domain volumes, as volume-increase was largely prevented when myonuclear accretion was inhibited. These findings seem paradoxical compared to the flexibility in myonuclear domains displayed during development in the present and accompanying study[40]. Clearly the experimental conditions are very different, and also in these experiments a reduction in nuclei led to reduced developmental growth and finally smaller fibers at the adult stage.

Some authors have suggested that accretion of satellite cells and an elevation of the number of myonuclei are not important for increasing cell size, but that satellite cells rather have the purposes of supporting muscle repair and extracellular matrix remodeling, and might not necessarily be a precursor to fusion for augmenting transcriptional capacity during adult muscle fiber hypertrophy[63].

In Drosophila larvae muscles nuclear size is related to cell volume[41], and we hypothesized that the observed flexibility in myonuclear domains revealed by the impairment of satellite cell fusion in developing muscle might be related to such a

mechanism, but nuclei were not found to be related to myonuclear domain size. While endoreplication occurs in Drosophila during maturation, this does not seem to be the case in vertebrates[64–66].

Like in the present study, previous studies in mammals have mainly related cell size to DNA content[41,46,49,67,68], and while this parameter is strongly correlated to nuclear size in a range of organisms[1,49,68–71], it has been suggested that the physical size of the nuclei might also be important. Thus, in cytokinesis-defective yeast, nuclear size correlates to cell size independent of DNA content[72], and the importance of nuclear size per se has also been discussed for cells in developing Xenopus embryos[73]. Although our data show that the DNA volume was unaltered, we cannot exclude that there are other qualitative differences between nuclei in small and large fibers.

In summary, we demonstrate that the number of nuclei in the muscle fiber syncytia adhere to fundamental scaling properties previously only reported for proliferative cells. The number of nuclei scaled sublinearly to cell volume, but close to linear with cell surface. We suggest that the sublinear scaling of nuclei to cellular volume might be a mechanism limiting fiber size.

## Methods

**Animal experiments.** A total of 14 female NMRI mice (postnatal day (P) 70–77) with a body weight of 28–34 g (31 ± 1.8 g, mean ± standard deviation) were used. Mice were housed in ambient temperature at 22 °C, with 45–55% humidity and 12/12 h dark/light cycle. The animal experiments were approved by the Norwegian Animal Research Authority and were conducted in accordance with the Norwegian Animal Welfare Act of 20th December 1974. The Norwegian Animal Research Authority provided governance to ensure that facilities and experiments were in accordance with the Act, National Regulations of 15th January 1996, and the European Convention for the Protection of Vertebrate Animals Used for Experimental and Other Scientific Purposes of 18th March 1986. For the developmental studies (see below), mice were housed in a room with an ambient temperature of 22 °C, with 30–75% humidity, and 10/14 h dark/light cycle and all animal procedures were approved by Cincinnati Children's Hospital Medical Center's Institutional Animal Care and Committee (see also Cramer et al. accompanying paper[40]).

**Denervation of the peroneal nerve innervating the EDL muscle in mice.** In eight mice, all age-matched to the control (six mice), a small incision laterally at the level of the knee was performed, the common peroneal nerve was exposed and cut, and its proximal end reflected and sutured to the subcutis to prevent reinnervation. After 14 days, the mice were prepared for single fiber imaging.

**In vivo injections and single cell imaging.** Before surgery, animals were deeply anesthetized by a single intraperitoneal injection of a ZRF cocktail (18.7 mg zolazepam, 18.7 mg tiletamine, 0.45 mg xylazine, and 2.6 mg fentanyl per ml) that were administered at a dose of 0.08 mL per 20 g body weight. The skin over the tibialis anterior was shaved and a small incision was made to expose the overlaying muscles that were subsequently retracted laterally to expose the EDL muscle. The lateral surface of the EDL contains virtually only type 2b fibers[13]. The epimysium was gently removed, and we took care not to damage the muscle. The exposed muscle was covered with a mouse Ringer's solution; NaCl 154 mM, KCl 5.6 mM, MgCl₂ 2.2 mM, and NaHCO₃ 2.4 mM, and held in place with a coverslip mounted approximately 2 mm above the muscle.

Animals were placed under a fixed-stage fluorescence microscope (Olympus BX50WI, Olympus, Japan) with a 20×, NA 0.3, long working distance water immersion objective. For in vivo labeling of nuclei and cytosol, single muscle cells in the EDL were injected with a solution containing 5′-TRITC or FITC-labeled random 17-mer oligonucleotide with a phosphorothioated backbone (Yorkshire Biosciences Ltd, Heslington, UK) dissolved in an injection buffer (10 mM NaCl, 10 mM Tris, pH 7.5, 0.1 mM EDTA, and 100 mM potassium gluconate) at a final concentration of 0.5 mM. Injections were made between the neuromuscular endplates and myotendinous junctions. The neuromuscular endplate were visualized by applying α-bungarotoxin conjugated to fluorescent dye (Molecular Probes) to the surface of the muscle for 2–3 min at a nonblocking concentration of 1 μg/mL[74].

After injection, the dye was given a sufficient time to diffuse (~30 min), allowing continuous fiber segments of about 0.5 mm (range: 0.29–0.83 mm) to be labeled, before we applied a solution of 4% paraformaldehyde (PFA) to the muscle surface to fixate the cell and the intracellular dye, thereby preserving cellular morphology and the cells natural length in the tissue. Animals were subsequently euthanized and the hindlimb were left in fixative until imaging.

**Analysis of single muscle cells from mice injected in vivo.** Muscle cells and nuclei in the EDL were imaged with a confocal microscope (Olympus FluoView 1000, BX61W1, Olympus, Japan) in optical sections, separated by z-axis steps of 1 μm to have the full three-dimensional data set of nuclei. Images (320 × 800 pixels × 1 μm voxel size) from optical sections of muscle cells were imported and analyzed for nuclear number, cellular volume and surface area using the Imaris Bitplane 8.3.1 software (Bitplane). Using the spot function in Imaris, a spot was automatically assigned to each nucleus based on the fluorescence intensity from the injected oligonucleotides, and if misaligned, manually repositioned. Volume and surface rendering were performed using the background fluorescence from the TRITC or FITC in the cytosol. Rendering of cellular geometry was performed using the fluorescence perimeter border of the cell in the cross-sectional direction as an outer limit, thereby preserving cell morphology during quantification. Importantly, end-plates and synaptic nuclei were excluded from further 3D rendering of the fiber, and only variables confined within the continuous fiber segment were extracted and analyzed.

**Preparation, imaging, and analysis of single muscle cells from human biopsies.** Images of single cells from the vastus lateralis muscle of human volunteers (males, age 20–29 years, and 67–85 kg body weight) were used with permission from ref. [12] and reanalyzed in this paper. The human subjects gave their written informed consent prior to participation in the study, and the study was approved by the Regional Ethics Committee of Stockholm, Sweden (DNR 2015/211-31/4), and was conducted in accordance with all relevant regulations and with the Declaration of Helsinki.

This muscle has an approximate equal distribution of type 1 and type 2 fibers[75], although the proportion of type 1 fibers is found to vary from 15 to 85% in a large cohort of individuals[76,77], whereas men have a greater ratio of type 2 to type 1 fiber mass[78].

Briefly, biopsies were fixed in 4% PFA in phosphate-buffered saline (pH 7.4). Single fibers were prepared by alkali maceration as previously described[45]. Single fibers in solution were then poured into a petri dish, placed on a glass slide (Superfrost plus, J1800AMNZ, Thermo Scientific), and mounted using DAPI Fluoromount G (Southern Biotech cat. 0100-20). Fiber segments were analyzed by acquiring images (640 × 640 pixels × 0.70 μm voxel size) on an Olympus BX61W1 upright confocal microscope, using Fluoview1000 with a 40× PlanApo (NA 0.80, Olympus) water immersion objective. A 405-nm laser was used to excite DAPI to visualize nuclei, and a 633-nm laser was used to visualize the fiber autofluorescence used for volume rendering. We ensured that each fiber segment was straight, and fibers that were hypercontracted or visually damaged were not included for further analysis. Acquired image stacks of the single fiber segments were reassembled to three-dimensional images using the Imaris Bitplane 8.3.1 software (Bitplane). The cell architecture was reconstructed and nuclei were counted automatically, then confirmed manually. A total of 267 muscle cells from seven individuals with 21–52 cells per biopsy were used. Segment lengths were limited by the ability to isolate long stretches of fiber, and the imaging was therefore standardized to imaging segments of 320 μm. The fibers were screened with differential interference contrast imaging and sarcomere lengths were averaged over ten sarcomeres for each fiber. There was minimal variability in length as measured by the muscle mean value, 2.05 ± 0.07 μm (SD, N = 7), which resulted in a relative SD of 3.4%. The variability between fibers within each muscle ranged from 3.3 to 6.5% (relative SD).

**Preparation and imaging of developmental cells from the EDL.** Isolating developing EDL muscles cells (predominately fast type 2x and 2b fibers[15,57,79,80]), with and without preventing satellite cell fusion by genetic deletion of myomaker at day 6 after birth (Δ2w), are described in detail in the accompanying paper[40]. In short, to ablate myomaker specifically in muscle satellite cells, Myomakerloxp/loxP mice were bred with mice carrying the muscle stem cell-specific Pax7CreER conditional Cre Recombinase. Myomakerlox P/loxP; Pax7CreER represent the experimental group (Δ2w), while Myomakerlox P/loxP serve as controls.

EDL myofibers stored in 1% paraformaldehyde were harvested in Dr. Millay's lab from 11 female and 8 male control mice, and from 13 female and 12 male experimental mice. In our lab, single fibers were placed on a glass slide (Superfrost Plus, Thermo Fisher Scientific) and the slides were mounted by a glass cover slip (No. 1.5, Marienfeld) with DAPI Fluoromount-G (Southern Biotech) to visualize nuclei. Slides were dried overnight and sealed with nail polish and stored no longer than 2 days before imaging.

Images were acquired with a 40× oil immersion objective (CPI, Plan Fluor, NA 1.3) on an Andor DragonFly (Andor, Oxford Instruments) confocal microscope with a Zyla4.2 sCMOS camera with a x–y resolution of 0.3 × 0.3 μm and z step size of 1 μm. Lasers with emission wavelength 405 (DAPI), 488, and 561 nm were used. Pixel binning of 2 × 2 was used to reduce time of image acquisition and improve signal-to-noise ratio. 25–40 cells from each muscle were analyzed. Nuclear number, DNA volume, and cellular volume were analyzed using the Imaris Bitplane 8.3.1 software (Bitplane) as described in previous sections. For measurements of the DNA/nuclear volume, based on the nuclear DAPI stain, the surface area detail level for the 3D rendering were set to 1 μm, with a diameter of largest sphere which fits into the object to be 5 μm for local intensity background subtraction.

**Power law and logarithms**. To investigate the scaling between nuclear number and size-related parameters, we used power-law relationships of the form:

$$y = ax^b,$$

which were linearized and analyzed in the logarithmic form,

$$\log_{10}(y) = \log_{10}(a) + b \times \log_{10}(x),$$

where $y$ and $x$ are variables, $a$ is the normalization constant and $b$ is the scaling exponent. An ordinary least square method (OLS) were used to fit log–log regression to data.

**Statistics**. Data were analyzed and plotted with Graphpad Prism 8 software. Data sets are presented as their arithmetic mean and their corresponding 95% CI if not stated otherwise. In order to statistically compare the scaling exponent from different groups of cells or from hypothetical values of $b$ we used the extra sum-of-squares $F$-test. For all remaining comparisons between group parameters at the level of cells or muscles we used a Welch's or nested $t$-test, Brown–Forsythe and Welch's ANOVA test to accommodate for the potential effect of dependency among samples and unequal sample sizes[81]. Specific statistical tests used are noted in individual figure legends and all inference is made upon the level of significance at a critical value of $\alpha = 0.05$.

**Reporting summary**. Further information on research design is available in the Nature Research Reporting Summary linked to this article.

## Data availability

The data that support the findings of this study are available as source data provided with this paper. Source data are provided with this paper.

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

## Acknowledgements

This work was supported by the Norwegian Research Council (grant: 240374) to K.G. E.E. was funded by the US-Norway Fulbright Foundation for Educational Exchange. Work in laboratory of D.P.M. was funded by the Cincinnati Children's Hospital Research Foundation, National Institutes of Health (R01AR068286, R01AG059605), and Pew Charitable Trusts.

## Author contributions

K-A.H., E.E., J.C.B., I.J., A.W.C., A.M-S., D.P.M., and K.G. designed experiments. K-A.H., E.E., and A.W.C. performed animal experiments. K-A.H. and E.E. designed and K-A.H. and I.J. performed image analysis. K-A.H. created figures. All authors analyzed the data and contributed to interpretations. K.G. and K-A.H. wrote the manuscript with input from all authors.

## Competing interests

The authors declare no competing interests.
