## [Peer Review File · Nature Communications]

Reviewers' Comments:

Reviewer #1:

Remarks to the Author:

This study shows that nuclear number scales with cross sectional area and volume of muscle fibers in mouse and humans. Specifically, this scaling relationship is sublinear, meaning that larger cells have proportionally fewer nuclei than smaller cells. Denervation of mouse fibers results in a decrease in cross-sectional area and muscle volume, but no change in nuclear numbers. This indicates that myonuclear domain sizes decrease. Inhibition of cell fusion results in smaller cells (cross section areas and volumes) and fewer nuclei. Under this experimental condition, domain sizes increase. In both manipulation scenarios, the absolute number of nuclei per cell affects the ability/flexibility of the cells to compensate: larger cells with more nuclei atrophy less and grow more than smaller cells with fewer nuclei.

While it is important to investigate size regulation and nuclear scaling in muscle fibers across species and conditions, this study provides very little new information. It is widely accepted that muscle fibers can grow by increasing the number of nuclei and/or increasing their cytoplasmic volume, i.e. the domain size per nucleus. The scaling relationship of nuclear number with cross sectional area is routinely used to assess muscle size. In vivo/in situ analysis and 3D reconstruction of mouse EDL muscles has been done previously (e.g. Bruusgaard et al 2003). To my knowledge, human muscle fibers have not been analyzed in 3D and the finding that there is similar scaling to mouse muscle fibers is exciting. The response of mouse muscle fibers to denervation or a limited pool of satellite cells is not new. The final conclusion that the ability of individual muscle fibers to compensate depends on the absolute number of nuclei per cell size is, no doubt, exciting. Nevertheless, the presentation and discussion of the results are confusing and often not clear for the reader. Cramer et al. also draw the same conclusion, but in a much more accessible way.

General comments/critiques on approach:

1. Muscle fiber types are known to have different scaling of nuclear number with fiber size. Was that considered in the choice of samples? This is not addressed nor discussed.
2. The mouse study was done with female mice and the human study with male individuals. Are there sex differences that should be considered?
3. Are mice at 10-11 weeks still growing? How does this stage compare to the age of the human biopsy volunteers? How does this compare to Stage P150?
4. Muscle fiber 3D analysis: which sections along the muscle fibers were analyzed? Was the NMJ or MTJ considered? What about the nuclei at the edges of the selected cell segments? For the injections to label nuclei, could satellite cell nuclei also take up the dye? If so, how are these eliminated from the nuclear count?
5. Muscle surface area: The discussion and suppl figure indicate that the number of nuclei is actually proportional with cell surface area. These data should be part of the main results. The 3D reconstructions should provide this information. But also data of this type were discussed in Bruusgaard et al. 2003.

Specific comments/critiques:

1. Highlights: These need to be rethought.
#2: the idea/fact that nuclear number determines muscle cell size is far from new.
#4: is this really new?
What about denervation results – why is this not a highlight?

2. Abstract

Line34: "scaling of muscle cells are poorly understood" ?

The abstract doesn't reflect the content of the paper very well, particularly as this lab has

published important work in this area.

3. Intro

Line112: "under-compensation": what about growth in this context? Wouldn't it make more sense to think about it as a growth potential. Small fibers, small domains; after fusion domain sizes increase until optimal nuclear scaling is established.

4. RESULTS

a. Number of nuclei scales sub-linearly to cell volume in mouse muscle fibers labelled in vivo

Line 121: "we visualized nuclei that stained intensely" – what does this mean? Does this mean that only nuclei on the top of the cell were considered?

Line 143: 'mice' instead of 'muscles'?

Line 150: "right skewed distribution" What does this indicate?

Line152: There is no plot comparing nuclear number and cell volume that also has a dashed line.

Line153: Please clarify how plot M relates to the statement "individual myonuclear domains increased with cell size".

Line 156: "was not linear" ? – Fig.1L looks linear to me.

Line 157: fractals- It is unclear what adding this statement means for the reader. Please clarify.

Figure1:

- Images and description on how nuclear parameters were quantified are very confusing

Brightness as selection criterium is not clear.

- Plots don't correspond well with what the text says.

b. Scaling behavior of adult human muscle are similar to mice in spite of differences in absolute numbers

Line 163: The EDL in mouse is a fast twitch muscle, whereas Is the m vastus lateralis is composed of both fast and slow-twitch fibers? Are these comparable then?

Line 165: "twice as large" figure shows cross sectional area (if there's multiple parameters that are used for assessing size. e.g. area and volume, it is important to be precise)

Lines191-195: These lines are unclear and the use of "compensation" here difficult to follow.

Figure2: Similar general issues as with Figure1

A-C: It appears that only one half of the fiber was analyzed. Where is the bottom?

F: distribution indicates that there are quite a few cells with a cross sectional area of 0 (zero) – Could the authors please clarify?

F,H,J: What are all the frequency/distribution plots for mouse and human actually telling us?

c. Myonuclei are not lost by denervation, but sub-linear scaling behavior is exaggerated by differential atrophy

Line 207: 'scaling exponent' - for scaling of what? This needs to be specified. Nuclear number with cell volume?

Line 208: 'smaller fibers' atrophied more – does this refer to 'cells with fewer nuclei at the time of denervation'? With multiple parameters for muscle size being quantified and shown, please be more precise as to which parameter small refers.

d. Scaling behavior is retained during post-natal growth, also after inhibiting myonuclear accretion genetically

Lines 230-237: It is not clear why only CSA size distributions are used for showing growth with and without satellite cells in Fig4A. Plots with direct comparison of CSA with volume and nuclear number at the different time points would be much more informative when investigating nuclear scaling.

Line 236: "radial growth": are the authors referring to the increase in cross-sectional area?

Line 237: "were much larger" : I assume that the authors are also referring to increased cross-sectional area.

Line 240: "nuclear density" indicates nuclear number per mm fiber. An increase means fusion in controls and no fusion in d2w mice. Does P150 = 21 weeks? What is the situation at week 10-11 (as analyzed in Fig.1)?

Line 243: 'steeper slope' : this is difficult to see in plot 4D

Lines 247onwards:

'scaling behavior' : please be precise, which scaling behavior (scaling of nuclear number with cell volume, it is assumed).

'was similar' – there was scaling but with a different normalizing constant; the slope of the linear correlation (scaling exponent) was similar.

e. Additional data questions:

Are nuclear sizes changing with changes in myonuclear domain sizes (as shown in other systems)? 3D reconstructions should provide this information.

Where scaling of nuclear numbers with muscle cell size is disrupted, are there changes in nuclear DNA content or other possible factors of compensation? (like translational, transcriptional differences?)

5. DISCUSSION

a. First paragraph:

Instead of large fibers 'under-compensating', what if they have just reached their perfect scaling relationship (of nuclear number with fiber size), and small fibers (while still growing) 'over-compensate' – i.e., have more nuclei per cell volume to promote growth. Please comment/clarify.

b. Lines 304-315: As mentioned earlier, the paper would be improved by integrating surface data into the results. Also, these data need to be discussed in the context of previous studies that link the number of nuclei and muscle surface area to nuclear positioning/distribution and thus the regulation of domain sizes (e.g. Bruusgaard et al. 2003 and Manhart et al., 2018).

c. Paragraph starting 320: this paragraph is very confusing as written but has the potential to really make the authors point. Please revise.

d. Paragraph starting 336: The use of the word 'upscaling' in this context: does it mean just growing, or does it refer to the size of the myonuclear domains (which is exactly what the MND hypothesis postulates)?

e.Paragraph starting 349: It is not clear what 'absolute ceiling' means; myonuclear domain sizes in d2w mice could be interpreted as showing nuclei at their maximum capacity, i.e. ceiling? It doesn't make sense to assume that normal muscle fibers under regular conditions would operate at max capacity, but instead optimize for low stress on the system (nice symmetrical gaussian distributions of size parameters). This would allow for the size flexibility that's been observed in so many different muscle studies.

f.Paragraph starting 377: This paragraph isn't clear, and the logic doesn't make sense. If DNA content regulates muscle cell size, and muscle cells are able increase DNA content via satellite cell fusion, it only makes sense if there is also a feedback mechanism where cell size regulates DNA content (i.e. the addition of new nuclei during growth).If the number of satellite cells is restricted in the adult, flexibility goes down. It doesn't seem so strange that during development, muscles are more flexible.

g.It has not been conclusively shown that resident myonuclei don't undergo replication.

h. Please reconsider the referencing as many, even from the senior author, appear left off throughout the manuscript.

Reviewer #2:

Remarks to the Author:

In the study entitled "Myonuclear content regulate cell size with similar scaling properties in mice, humans and after postnatal reduction of myonuclear number" Hansson and co-workers have studied how myonuclei scale to muscle fiber size in rodents, humans and genetically manipulated mice with reduced myonuclear number. It is shown that myonuclei number scale sub-linearly to cell volume and more closely to muscle cell surface. It is speculated that long-term regulation of muscle fiber size may be related to surface receptors alternatively to diffusion properties over the cell membrane.

General comments:

This is a well written manuscript by an established research group that has made significant contributions to our understanding of myonuclear organization and also developed a powerful rodent model for in vivo imaging of myonuclei.

Specific comments:

p. 19 1st para It is stated that the larger myonuclear domains in larger muscle cells might be a factor limiting muscle fiber hypertrophy. How does this fit with "double muscle" myostatin knock out mice with large muscle volumes in muscles with very large myonuclear domains?

p. 27 It is stated that hypercontracted fibers were discarded, but there is no information on sarcomere length (SL) monitoring in the human muscle fiber segments. How was SL monitored between different fibers and which range of SLs were included in the study?

Minor comments:

p. 13 l.203-204 14 d duration of denervation is short. Add information on myonuclear number after longer periods of denervation, i.e., 3 months or longer.

p. 16 l.253 ...groups....

We are grateful to the reviewers for doing such a thorough job to improve our manuscript.

General rebuttal comments

Reviewer number 1 seems to disagree with some of our interpretations, namely that scaling properties itself could explain cell size determination rather than “purpose built” regulatory pathways, this is a major point in our paper, and we think our viewpoint is a valid interpretation (see below).

The reviewer also seems to suggest that domains are in general very flexible, in our opinion as a general rule this goes against most mammalian literature as we interpret it, at least not without a functional “penalty” (discussed below).

We thank the reviewer for suggesting that the nuclear size could play a role. It prompted us to make new measurements and we have added a new figure addressing this point. We have also included more references related to this point.

Reviewer 1 felt that the paper was not clearly written, and we have rewritten the MS extensively in order to make our arguments clearer. We hope this makes the paper better, even if the reviewer might still not share all our interpretations.

We respond in black to the specific points made by the reviewers in green.

Reviewer #1 (Remarks to the Author):

This study shows that nuclear number scales with cross sectional area and volume of muscle fibers in mouse and humans. Specifically, this scaling relationship is sublinear, meaning that larger cells have proportionally fewer nuclei than smaller cells. Denervation of mouse fibers results in a decrease in cross-sectional area and muscle volume, but no change in nuclear numbers. This indicates that myonuclear domain sizes decrease. Inhibition of cell fusion results in smaller cells (cross section areas and volumes) and fewer nuclei. Under this experimental condition, domain sizes increase. In both manipulation scenarios, the absolute number of nuclei per cell affects the ability/flexibility of the cells to compensate: larger cells with more nuclei atrophy less and grow more than smaller cells with fewer nuclei.

While it is important to investigate size regulation and nuclear scaling in muscle fibers across species and conditions, this study provides very little new information. It is widely accepted that muscle fibers can grow by increasing the number of nuclei and/or increasing their cytoplasmic volume, i.e. the domain size per nucleus. The scaling relationship of nuclear number with cross sectional area is routinely used to assess muscle size. In vivo/in situ analysis and 3D reconstruction of mouse EDL muscles has been done previously (e.g. Bruusgaard et al 2003). To my knowledge, human muscle fibers have not been analyzed in 3D and the finding that there is similar scaling to mouse muscle fibers is exciting. The response of mouse muscle fibers to denervation or a limited pool of satellite cells is not new. The final conclusion that the ability of individual muscle fibers to compensate depends on the absolute number of nuclei per cell size is, no doubt, exciting. Nevertheless, the

presentation and discussion of the results are confusing and often not clear for the reader. Cramer et al. also draw the same conclusion, but in a much more accessible way.

The important point for us is that all the experimental models in this paper are analyzed with precise confocal imaging and 3D reconstruction techniques, and that they are all put into the theoretical framework of scaling, and that this analysis reveals general principles across models and species.

While the reviewer show enthusiasm for our data on human material and the titration of nuclei using the genetic model, we would also like to highlight the importance of the normal and denervated mouse data. Previous material on normal mice (Bruusgaard et al 2003) was based on a crude estimate of fiber size, and non-confocal imaging, and was not related to general scaling theory. To our knowledge the present paper is the first where confocal imaging is used on normal or denervated muscles in vivo, and to our knowledge 3D reconstruction and scaling considerations have not been applied to denervated fibers before.

General comments/critiques on approach:

1. Muscle fiber types are known to have different scaling of nuclear number with fiber size. Was that considered in the choice of samples? This is not addressed nor discussed.

The in vivo imaging techniques limits the number of muscles available, and our techniques does not allow for fiber typing of the reconstructed fibers. But the injected surface fibers at the lateral side of the EDL are almost exclusively type IIB. The human biopsy material from vastus lateralis contains a mixture of type I and type II fibers as we have addressed in a previous paper (Psilander et al, 2019). We have added text describing the fiber type distribution in the methods. To us the heterogeneity is, however, not a hurdle, and we find it interesting that the scaling properties seemed to apply even for heterogenous materials.

2. The mouse study was done with female mice and the human study with male individuals. Are there sex differences that should be considered?

We would think that the species differences are more important than sex, but again the same scaling principles seems to apply across models in spite of the heterogeneity.

3. Are mice at 10-11 weeks still growing? How does this stage compare to the age of the human biopsy volunteers? How does this compare to Stage P150?

Our normal mouse material is P70-77 (10-11 weeks), and mouse growth curves flattens out around P50. Some weight gain due to fat takes place more or less throughout the lifespan. It is hard to compare age in rodents and humans, but the all the material except the early developmental timepoints could be considered “young adults”. We now give the age as P-days in all places to aid readability.

4. Muscle fiber 3D analysis: which sections along the muscle fibers were analyzed ? Was the NMJ or MTJ considered? What about the nuclei at the edges of the selected cell segments?

For the injections to label nuclei, could satellite cell nuclei also take up the dye? If so, how are these eliminated from the nuclear count?

These fibers are rather long (mm scale) so segments were analyzed. They are far from the MTJ, we labelled the NMJ with alpha bungarotoxin, and they were excluded. They are also in most cases discernable as a cluster of more roundish nuclei. We apologize for this not being included in the methods; it is now included. We also now make clear that the segments were rather far from the MTJ so they were not included.

5. Muscle surface area: The discussion and suppl figure indicate that the number of nuclei is actually proportional with cell surface area. These data should be part of the main results. The 3D reconstructions should provide this information. But also, data of this type were discussed in Bruusgaard et al. 2003.

We have now added data related to myonuclear domains as surface domains in all relevant figures.

Specific comments/critiques:

1. Highlights: These need to be rethought.

#2: the idea/fact that nuclear number determines muscle cell size is far from new.

#4: is this really new?

What about denervation results – why is this not a highlight?

Nat. Com. does not require highlights so this has section has been omitted.

2. Abstract

Line34: “scaling of muscle cells are poorly understood” ?

The abstract doesn’t reflect the content of the paper very well, particularly as this lab has published important work in this area.

The abstract has been completely rewritten, but (unfortunately) also shortened considerably to fit Nat. Com. length restrictions.

3. Intro

Line112: “under-compensation”: what about growth in this context? Wouldn’t it make more sense to think about it as a growth potential. Small fibers, small domains; after fusion domain sizes increase until optimal nuclear scaling is established.

We agree with the reviewer that our phrasing was ambiguous, and have rewritten the paragraph and are not any longer using the term “under-compensation”. What we meant was that it was an under-compensation if the number were to maintaining a constant myonuclear domain volumes when going from small to large fibers. A hypothetical reserve capacity for growth is a different issue. As we interpret the literature there is very little reserve capacity for growth without adding new nuclei in most models (Eger et al Development 2016; Goh et al. eLife 2017; Goh et al. eLife 2019; see also discussion in Eger

et al. Development 2017 and Eftestøl JAP 2020).

4. RESULTS

a. Number of nuclei scales sub-linearly to cell volume in mouse muscle fibers labelled in vivo

Line 121: “we visualized nuclei that stained intensely” – what does this mean? Does this mean that only nuclei on the top of the cell were considered?

No, and we have rephrased in order to make it clearer: “By 3D confocal imaging, we visualized nuclei that were sharply delineated with intense labelling, as well as cell geometry based on a diffuse relatively faint background staining of the cytosol (Fig. 1 A)” It is correct that nuclei deeper in the stack were fainter due to laser penetration, but we don't consider identifying nuclei much of a problem when going through individual stacks, all nuclei were of course considered, not only those at the top.

As seen in Fig.

Line 143: ‘mice’ instead of ‘muscles’?

Changed

Line 150: “right skewed distribution” What does this indicate?

It is just meant as a description of the data, the distribution is not symmetrical (e.g. normal).

Line152: There is no plot comparing nuclear number and cell volume that also has a dashed line.

We have re-phrased this, and the figures have been altered

Line153: Please clarify how plot M relates to the statement “individual myonuclear domains increased with cell size”.

Line 156: “was not linear” ? – Fig.1L looks linear to me.

Yes, but it is a log-log plot (Fig.1o) , and a straight line in log-log space doesn't indicate a linear relationship. $b=0.73$ translates into $N \propto V^{0.73}$, and thereby a sub-linear relationship.

Line 157: fractals- It is unclear what adding this statement means for the reader. Please clarify.

We agree this was confusing and have taken out this part.

Figure1:

- Images and description on how nuclear parameters were quantified are very confusing
- Brightness as selection criterium is not clear.
- Plots don't correspond well with what the text says.

b. Scaling behavior of adult human muscle are similar to mice in spite of differences in absolute numbers

Line 163: The EDL in mouse is a fast twitch muscle, whereas is the m vastus lateralis is composed of both fast and slow-twitch fibers? Are these comparable then?

To us it is interesting that in spite of differences in species, gender and fiber-type the different experimental materials share common scaling properties.

Line 165: "twice as large" figure shows cross sectional area (if there's multiple parameters that are used for assessing size. e.g. area and volume, it is important to be precise)
We agree and have rewritten this part

Lines 191-195: These lines are unclear and the use of "compensation" here difficult to follow.

We agree and have rewritten this part

Figure 2: Similar general issues as with Figure 1

This figure has been redesigned as well.

A-C: It appears that only one half of the fiber was analyzed. Where is the bottom?

It is indeed a whole fiber segment lying on microscope slides and the shape is distorted compared to the in situ situation. Such shape-distortions are commonly seen in this type of preparations. We have included more explanation in the text. For human material in situ experiments are not feasible.

F: distribution indicates that there are quite a few cells with a cross sectional area of 0 (zero) – Could the authors please clarify?

F,H,J: What are all the frequency/distribution plots for mouse and human actually telling us?

We thank the reviewer for pointing this mistake out, it appears that we used the wrong non-linear function and we have corrected this. We also agree that perhaps just to presented a fitted mathematical distribution is not very informative. We have therefore substituted all these figures in the manuscript with data histograms as well as the correct mathematical function superimposed. We feel that the distribution of all of the actual data from each fiber is informative in addition to averages of each muscle.

c. Myonuclei are not lost by denervation, but sub-linear scaling behavior is exaggerated by differential atrophy

Line 207: 'scaling exponent' - for scaling of what? This needs to be specified. Nuclear number with cell volume?

Now specified

Line 208: 'smaller fibers' atrophied more – does this refer to 'cells with fewer nuclei at the time of denervation'? With multiple parameters for muscle size being quantified and shown, please be more precise as to which parameter small refers.

We have rephrased this part.

d. Scaling behavior is retained during post-natal growth, also after inhibiting myonuclear accretion genetically

Lines 230-237: It is not clear why only CSA size distributions are used for showing growth with and without satellite cells in Fig4A. Plots with direct comparison of CSA with volume and nuclear number at the different time points would be much more informative when investigating nuclear scaling.

We don't understand what the reviewer means here. CSA times length equals volume, and when the fiber segments was standardized CSA essentially equals volume. We have plotted nuclear number against volume in Fig. 5a while showing CSA as a function of developmental time explicitly in Fig.4a and b, as suggested by the reviewer.

Line 236: "radial growth": are the authors referring to the increase in cross-sectional area?

Yes, we use this term for the developmental data to be precise, since during development there is also a growth in length that we are not referring to here.

Line 237: "were much larger" : I assume that the authors are also referring to increased cross-sectional area.

Yes, and we specify that now.

Line 240: "nuclear density" indicates nuclear number per mm fiber. An increase means fusion in controls and no fusion in d2w mice. Does P150 = 21 weeks? What is the situation at week 10-11 (as analyzed in Fig.1)?

P150 is approximately 21 weeks. 10-11 weeks would be P70-77. We don't know exactly when altered scaling behavior related to the nuclear reduction sets in, but both P42 and P77 would be after the major growth phase in normal animals

Line 243: 'steeper slope' : this is difficult to see in plot 4D

We thank the reviewer for pointing out that this sentence was misleading, and have rephrased it

Lines 247onwards:

'scaling behavior' : please be precise, which scaling behavior (scaling of nuclear number with cell volume, it is assumed).

'was similar' – there was scaling but with a different normalizing constant; the slope of the linear correlation (scaling exponent) was similar.

We have done extensive rewriting in order to improve clarity

e. Additional data questions:

Are nuclear sizes changing with changes in myonuclear domain sizes (as shown in other systems)? 3D reconstructions should provide this information.

We are grateful for the suggestion, and have performed the analysis, included a new data figure and appropriate text in results and discussion.

Where scaling of nuclear numbers with muscle cell size is disrupted, are there changes in nuclear DNA content or other possible factors of compensation? (like translational, transcriptional differences?)

This is beyond the scope of this paper. It is not known what the “bottleneck” is for the requirement of nuclei. In general, it is believed that cell size is dependent on the balance between protein synthesis and proteolysis, nobody has to our knowledge been able to give a full account for such a balance.

5. DISCUSSION

a. First paragraph:

Instead of large fibers ‘under-compensating’, what if they have just reached their perfect scaling relationship (of nuclear number with fiber size), and small fibers (while still growing) ‘over-compensate’ – i.e., have more nuclei per cell volume to promote growth. Please comment/clarify.

We have rewritten the paragraph in order to make it clearer. The classical domain hypothesis is related to volume, and in that sense the volume is undercompensated for in all the different materials in this paper when going from small to large fibers. As we see it, it is also clear that in adult fibers no or little CSA growth will occur unless you increase the number of nuclei, the number of nuclei seems to be a limiting factor. Myomaker knock outs and other models have proven this (see discussion in Eftestøl et al. JAP 2020, and Eger et al Development 2016; Goh et al. eLife 2017; Goh et al. eLife 2019; see also discussion in Eger et al. Development 2017). To us the interesting observation is that this “under-compensation” for increasing volume is universal across the various experimental groups growing and not growing, the muscle fibers become more DNA scarce the larger they get, and muscle is very low in DNA content compared to other cells. Scaling theory generally aims at showing such universal relationships.

b. Lines 304-315: As mentioned earlier, the paper would be improved by integrating surface data into the results. Also, these data need to be discussed in the context of previous studies that link the number of nuclei and muscle surface area to nuclear positioning/distribution and thus the regulation of domain sizes (e.g. Bruusgaard et al. 2003 and Manhart et al., 2018).

While we have included more data and text on scaling to surface, e.g. the Manhart reference, distribution and regulation of nuclear positioning is outside the scope of this paper. Changes in distribution would not influence the calculated domain sizes, and the observed scaling properties.

c. Paragraph starting 320: this paragraph is very confusing as written but has the potential to really make the authors point. Please revise.

We have revised the text and added a discussion on the possibility of polyploidy in this material, and related to the new measurements we have added.

d. Paragraph starting 336: The use of the word 'upscaling' in this context: does it mean just growing, or does it refer to the size of the myonuclear domains (which is exactly what the MND hypothesis postulates)?

It is a theoretical up-scaling according to the scaling principles that we and others have uncovered. It does not mean growth. We have made changes in the text that we hope makes it clearer.

e. Paragraph starting 349: It is not clear what 'absolute ceiling' means; myonuclear domain sizes in d2w mice could be interpreted as showing nuclei at their maximum capacity, i.e. ceiling? It doesn't make sense to assume that normal muscle fibers under regular conditions would operate at max capacity, but instead optimize for low stress on the system (nice symmetrical gaussian distributions of size parameters). This would allow for the size flexibility that's been observed in so many different muscle studies.

Here our opinion diverges from the reviewer and we don't understand what the reviewer is referring to by stating: "*This would allow for the size flexibility that's been observed in so many different muscle studies.*", as no references are provided. As we interpret the current mammalian literature there is rather little flexibility demonstrated. For example, during overload-induced hypertrophy, recruitment of myonuclei precedes growth (see Bruusgaard 2010, PNAS), and as discussed in the MS hypertrophy is largely prevented when myonuclear accretion is prevented. Although non physiological molecular manipulations (e.g. myostatin blocking) display large fibers with few nuclei, they have impaired function with reduced specific force, reduced myofibrillar apparatus, and they even do not taste good (Belgian blue). The flexibility during developmental growth observed here and in the accompanying paper is discussed in both papers. Please note that our Discussion has been extensively rewritten.

f. Paragraph starting 377: This paragraph isn't clear, and the logic doesn't make sense. If DNA content regulates muscle cell size, and muscle cells are able increase DNA content via satellite cell fusion, it only makes sense if there is also a feedback mechanism where cell size regulates DNA content (i.e. the addition of new nuclei during growth). If the number of satellite cells is restricted in the adult, flexibility goes down. It doesn't seem so strange that during development, muscles are more flexible.

We have rewritten the text in order to improve clarity. But we would like to stress that the sublinear scaling itself might limit fiber size without any feedback mechanism. This is actually an important point of the paper, that scaling alone could limit the growth by DNA getting to scarce.

g. It has not been conclusively shown that resident myonuclei don't undergo replication.

We have changed the wording, and have added more references to prove our point. Epistemologically, it is hard to prove that a phenomenon never exists. As we interpret existing literature on vertebrates there is no thymidine incorporation in muscle undergoing growth. And our own attempts to show EdU incorporation during hypertrophic growth came out negative (unpublished). Finally, it is in agreement with the new data we have included on request of the reviewer (Fig. 6b-f).

h. Please reconsider the referencing as many, even from the senior author, appear left off throughout the manuscript.

We have added more text-references

Reviewer #2 (Remarks to the Author):

In the study entitled "Myonuclear content regulate cell size with similar scaling properties in mice, humans and after postnatal reduction of myonuclear number" Hansson and co-workers have studied how myonuclei scale to muscle fiber size in rodents, humans and genetically manipulated mice with reduced myonuclear number. It is shown that myonuclei number scale sub-linearly to cell volume and more closely to muscle cell surface. It is speculated that long-term regulation of muscle fiber size may be related to surface receptors alternatively to diffusion properties over the cell membrane.

General comments:

This is a well written manuscript by an established research group that has made significant contributions to our understanding of myonuclear organization and also developed a powerful rodent model for in vivo imaging of myonuclei.

Specific comments:

p. 19 1st para It is stated that the larger myonuclear domains in larger muscle cells might be a factor limiting muscle fiber hypertrophy. How does this fit with "double muscle" myostatin knock out mice with large muscle volumes in muscles with very large myonuclear domains?

We refer to this (Omairi, et al., 2016; Matsakas, et al., 2012; Metzger, et al., 2012; Bruusgaard, et al., 2005). Myostatin muscles have low specific force, are scarcer in sarcomeres, and even does not taste that good.

p. 27 It is stated that hypercontracted fibers were discarded, but there is no information on sarcomere length (SL) monitoring in the human muscle fiber segments. How was SL monitored between different fibers and which range of SLs were included in the study?

We apologize for not having included this. The material was prescreened with DIC (differential interference contrast) to visualize the sarcomeres during our work with the previous paper Psilander et al. 2019, and fibers with contractures were discarded. The sarcomere lengths are now given in the method part.

Minor comments:

p. 13 l.203-204 14 d duration of denervation is short. Add information on myonuclear number after longer periods of denervation, i.e., 3 months or longer.

Injections are not feasible in long term denervated muscles, we have however included a reference to Wada et al. 2002 who saw no loss of myonuclei after 4 months of denervation in isolated fiber segments.

p. 16 l.253 ...groups....

Corrected.

Reviewers' Comments:

Reviewer #1:

Remarks to the Author:

This is a re-review of a Hansson et al.

As in the first review, the authors describe their approach to investigate nuclear scaling in the context of the muscle fibers. No doubt, this is a very interesting area and will have significant impact for the muscle field. The authors have responded to the initial critiques and have changed the text a great deal, adding clarification of their experimental approaches, as well as additional data.

Concerns remain with the response to the initial review, new data, and the text itself. Some of these are minor and some major. These are listed below as a comments to the authors response to review.

1-Initial comment: . Muscle fiber types are known to have different scaling of nuclear number with fiber size. Was that considered in the choice of samples? This is not addressed nor discussed.

Authors' Response: The in vivo imaging techniques limits the number of muscles available, and our techniques does not allow for fiber typing of the reconstructed fibers. But the injected surface fibers at the lateral side of the EDL are almost exclusively type IIB. The human biopsy material from vastus lateralis contains a mixture of type I and type II fibers as we have addressed in a previous paper (Psilander et al, 2019). We have added text describing the fiber type distribution in the methods. To us the heterogeneity is, however, not a hurdle, and we find it interesting that the scaling properties seemed to apply even for heterogenous materials."

New comment: While it is indeed interesting that the scaling properties seem to apply, papers in the literature suggest that type I and type II fibers differ in myonuclear number (eg Qaiser and Larsson, 2014) and in myonuclear accretion during hypertrophy induced by training in humans (e.g Fry CS et al., J Physiol 2014 and others listed in this review: Murach et al., Physiology 2018).

Hence it seems that the types of samples used should be clearly described so that others can rigorously test and build upon what was described in this manuscript. Moreover the authors should discuss these data in light of their own.

2- Initial Comment 2: The mouse study was done with female mice and the human study with male individuals. Are there sex differences that should be considered?

Authors' Response: We would think that the species differences are more important than sex, but again the same scaling principles seems to apply across models in spite of the heterogeneity.

New comment: Fiber type composition and size are different between males and females (Haizlip et al., Physiology,2015). Therefore, some discussion of this is relevant for scientific rigor and for others to confirm and build upon. It should be stated explicitly in the text the sex and age of the human fibers, in line 147 and for the mice in line 97.

In addition, it should be clear how many fibers were assessed per individual mouse and human. It was difficult to find this information, particularly in light that some data points were removed from the human data set. It is not clear why these data were thrown out and if keeping it in would have significantly altered the dataset. If that dataset was kept in or analyzed separately, how would it compare to the mouse data?

3- Initial comment 4. Muscle fiber 3D analysis: which sections along the muscle fibers were analyzed? Was the NMJ or MTJ considered? What about the nuclei at the edges of the selected cell segments? For the injections to label nuclei, could satellite cell nuclei also take up the dye? If so, how are these eliminated from the nuclear count?

Authors' Response: These fibers are rather long (mm scale) so segments were analyzed. They are far from the MTJ, we labelled the NMJ with alpha bungarotoxin, and they were excluded. They are also in most cases discernable as a cluster of more roundish nuclei. We apologize for this not being included in the methods; it is now included. We also now make clear that the segments were rather far from the MTJ so they were not included.

New Comment: The additional information is welcomed, thank you! However, it is unclear if one is determining the myonuclear domains throughout the muscle, why one would remove the NMJ nuclei. Why wouldn't these be taken into consideration when assessing the scaling in the myofiber? Also, please comment on whether satellite cells could be labeled by this technique.

4- Initial comment: Are nuclear sizes changing with changes in myonuclear domain sizes (as shown in other systems)? 3D reconstructions should provide this information.

Authors' response: We are grateful for the suggestion, and have performed the analysis, included a new data figure and appropriate text in results and discussion.

New comment: There are unfortunately issues with the analysis and the conclusions. Firstly, Nuclear size cannot be accurately measured by Dapi. Dapi measures DNA. To measure Nuclear size, a marker like Lamin needs to be used. Secondly, depending on compaction/chromatin state/other nuclear compartments/cell state, the size of the nucleus can change. While correlations have been found between DNA content and nuclear size, nuclear size varies with cell size (Neumann and Nurse, 2007). Note that DNA content appears to set the lower limit of nuclear size (see reviews by Levy lab). How does total nuclear size scale with cell size (only myonuclear domain size was shown)?

5- Initial Comment e.Paragraph starting 349: It is not clear what 'absolute ceiling' means; myonuclear domain sizes in d2w mice could be interpreted as showing nuclei at their maximum capacity, i.e. ceiling? It doesn't make sense to assume that normal muscle fibers under regular conditions would operate at max capacity, but instead optimize for low stress on the system (nice symmetrical gaussian distributions of size parameters). This would allow for the size flexibility that's been observed in so many different muscle studies.

Authors' Response: Here our opinion diverges from the reviewer and we don't understand what the reviewer is referring to by stating: "This would allow for the size flexibility that's been observed in so many different muscle studies.", as no references are provided. As we interpret the current mammalian literature there is rather little flexibility demonstrated. For example, during overload-induced hypertrophy, recruitment of myonuclei precedes growth (see Bruusgaard 2010, PNAS), and as discussed in the MS hypertrophy is largely prevented when myonuclear accretion is prevented. Although non physiological molecular manipulations (e.g. myostatin blocking) display large fibers with few nuclei, they have impaired function with reduced specific force, reduced myofibrillar apparatus, and they even do not taste good (Belgian blue). The flexibility during developmental growth observed here and in the accompanying paper is discussed in both papers. Please note that our Discussion has been extensively rewritten.

New Comment: The current mammalian literature that was being referred to include the review by Murach et al., 2018 and many references therein, including Kadi et al., 2004, Verney et al., 2008, and Verdijk et al., 2009. Perhaps these should be discussed?

Please check through the text for grammar and clarity
As examples:

Line 105 "was ranging" should be ranged

Line 106 "displaying" should be displayed

Line 372 "The exceptions to similarity in scaling exponents, were..."

Line 380-81 "...a cytoarchitectural reminiscence of cell size ..."

Reviewer #2:

Remarks to the Author:

Reviewer #2

I find the response to my comment regarding myostatin -/- mice surprising. These mice have very large muscle fiber sizes in spite of proportionally low number of myonuclei. This does not fit with the notion that myonuclei number is a general limiter of fiber growth. The muscle fibers of these mice are strong but not in proportion to size resulting in lower specific tensions. However, this study does not include information on regulation of muscle contraction or contractile protein content in any of the other muscles included in this study and is solely focusing on myonuclear number vs. fiber size. The authors comment regarding the taste of myostatin deficient muscles is irrelevant

The initial comment from the reviewer in the first report is given in green
Our initial response is given in blue
The reviewer's new response in the second round is given in bold
Our latest response is given in red

Reviewer #1

Reviewer number 1 seems to disagree with some of our interpretations, namely that scaling properties itself could explain cell size determination rather than "purpose built" regulatory pathways, this is a major point in our paper, and we think our viewpoint is a valid interpretation (see below).

The reviewer also seems to suggest that domains are in general very flexible, in our opinion as a general rule this goes against most mammalian literature as we interpret it, at least not without a functional "penalty" (discussed below).

We thank the reviewer for suggesting that the nuclear size could play a role. It prompted us to make new measurements and we have added a new figure addressing this point. We have also included more references related to this point.

Reviewer 1 felt that the paper was not clearly written, and we have rewritten the MS extensively in order to make our arguments clearer. We hope this makes the paper better, even if the reviewer might still not share all our interpretations.

This is a re-review of a Hansson et al.

As in the first review, the authors describe their approach to investigate nuclear scaling in the context of the muscle fibers. No doubt, this is a very interesting area and will have significant impact for the muscle field. The authors have responded to the initial critiques and have changed the text a great deal, adding clarification of their experimental approaches, as well as additional data.

Concerns remain with the response to the initial review, new data, and the text itself. Some of these are minor and some major. These are listed below as a comments to the authors response to review.

1-Initial comment: Muscle fiber types are known to have different scaling of nuclear number with fiber size. Was that considered in the choice of samples? This is not addressed nor discussed.

Authors' Response: The in vivo imaging techniques limits the number of muscles available, and our techniques does not allow for fiber typing of the reconstructed fibers. But the injected surface fibers at the lateral side of the EDL are almost exclusively type IIB. The human biopsy material from vastus lateralis contains a mixture of type I and type II fibers as we have addressed in a previous paper (Psilander et al, 2019). We have added text describing the fiber type distribution in the methods. To us the heterogeneity is, however, not a hurdle, and we find it interesting that the scaling properties seemed to apply even for heterogenous materials."

New comment: While it is indeed interesting that the scaling properties seem to apply, papers in the literature suggest that type I and type II fibers differ in myonuclear number (eg Kaiser and Larsson, 2014) and in myonuclear accretion during hypertrophy induced by training in humans (e.g Fry CS et al., J Physiol 2014 and others listed in this review: Murach et al., Physiology 2018).

Hence it seems that the types of samples used should be clearly described so that others can rigorously test and build upon what was described in this manuscript. Moreover the authors should discuss these data in light of their own.

Response 1: We describe the material in the method section with respect to age, fiber type, species and sex so people can evaluate our data with respect to these parameters (Lines 509-510, 529-530, 564-568 and 591). We have added a paragraph discussing the matter as suggested by the reviewer (Lines 391-399). The new paragraph hopefully makes this clearer to the readership, but a detailed

discussion about the many and complicated differences in biology of different muscles etc. we consider outside the scope of our paper since our data points to a common underlying principle. In more technical terms there was a strong predictive relationship between nuclear number and fiber size, which is unlikely to come about stochastically. In fact, given the 95% CI's for the mice in vivo data and the human data the coefficient of determination (R^2) is about 80% and 63%, respectively, rendering these relationships with amazingly high predictive power.

2- Initial Comment 2: The mouse study was done with female mice and the human study with male individuals. Are there sex differences that should be considered?

Authors' Response: We would think that the species differences are more important than sex, but again the same scaling principles seems to apply across models in spite of the heterogeneity.

New comment: Fiber type composition and size are different between males and females (Haizlip et al., Physiology,2015). Therefore, some discussion of this is relevant for scientific rigor and for others to confirm and build upon. It should be stated explicitly in the text the sex and age of the human fibers, in line 147 and for the mice in line 97.

Response 2a: We have included the sex and age in what was line 147 and in line 97 for clarity, it is also given in methods.

In addition, it should be clear how many fibers were assessed per individual mouse and human. It was difficult to find this information, particularly in light that some data points were removed from the human data set. It is not clear why these data were thrown out and if keeping it in would have significantly altered the dataset. If that dataset was kept in or analyzed separately, how would it compare to the mouse data?

Response 2b: Number of fibers assessed per individual is given in the Results and Methods section, see e.g. lines 97, 509, 519, 582, 603-604. And also, in the figure legends of figure 1-5.

Importantly, NO data points were removed from the dataset, $n=267$ in the initial nor the revised manuscript. If the comment concerns (line175-177 in previous submission) this was merely a paper-exercise to see IF removing two subjects to achieve a unimodal distribution, and explain the bimodal distribution. We stress that the graph includes all subjects ($N=7$) and cells ($n=267$). We apologize for not writing this clearer, and have changed the phrasing (lines 181-183).

3- Initial comment 4. Muscle fiber 3D analysis: which sections along the muscle fibers were analyzed ? Was the NMJ or MTJ considered? What about the nuclei at the edges of the selected cell segments? For the injections to label nuclei, could satellite cell nuclei also take up the dye? If so, how are these eliminated from the nuclear count?

Authors' Response: These fibers are rather long (mm scale) so segments were analyzed. They are far from the MTJ, we labelled the NMJ with alpha bungarotoxin, and they were excluded. They are also in most cases discernable as a cluster of more roundish nuclei. We apologize for this not being included in the methods; it is now included. We also now make clear that the segments were rather far from the MTJ so they were not included.

New Comment: The additional information is welcomed, thank you! However, it is unclear if one is determining the myonuclear domains throughout the muscle, why one would remove the NMJ nuclei. Why wouldn't these be taken into consideration when assessing the scaling in the myofiber?

Response 3a: We have not removed the synaptic nuclei but selected segments not containing synaptic nuclei for our analysis. These nuclei have a unique transcriptional profile and their gene

products are localized to the synapse, and these nuclei probably don't contribute much to producing non-synaptic-proteins. Synaptic nuclei constitute about 1% of the nuclei in a fiber, but including them in a segment might introduce spread in the counting, and we wanted to avoid this complication. We have included a new paragraph in the results to explain these issues (Lines 98-104).

Also, please comment on whether satellite cells could be labeled by this technique.

Response 3b: Satellite cells: Indeed, it is a major strength of the injection (the water-soluble label is not taken up, but injected with a micropipette) methods that it labels myonuclei confined within one sarcolemma, and does not label satellite cells (see Bruusgaard et al 2003). We apologize for having overlooked this question in the previous re-submission, and we have included a statement of this issue with a reference to Bruusgaard 2003 (line 99).

4- Initial comment: Are nuclear sizes changing with changes in myonuclear domain sizes (as shown in other systems)? 3D reconstructions should provide this information.

Authors' response: We are grateful for the suggestion, and have performed the analysis, included a new data figure and appropriate text in results and discussion.

New comment: There are unfortunately issues with the analysis and the conclusions. Firstly, Nuclear size cannot be accurately measured by Dapi. Dapi measures DNA. To measure Nuclear size, a marker like Lamin needs to be used. Secondly, depending on compaction/chromatin state/other nuclear compartments/cell state, the size of the nucleus can change. While correlations have been found between DNA content and nuclear size, nuclear size varies with cell size (Neumann and Nurse, 2007). Note that DNA content appears to set the lower limit of nuclear size (see reviews by Levy lab). How does total nuclear size scale with cell size (only myonuclear domain size was shown)?

Response 4a: We have performed the analysis on our 3D reconstructions as requested by the reviewer after the previous submission. The reviewer is technically right that we are not measuring nuclear size, but DNA volume is closer to the issues we are discussing in the paper. Most literature on the animal kingdom has focused on DNA content rather than nuclear size per se, and nuclear size has been used as a proxy for DNA content in most cases. In animals, DNA content and nuclear size seems to be well correlated, in particular in post mitotic cells. The reviewer is, however, correct that in yeast under special experimental conditions related to manipulated mitosis, cell size was correlated to nuclear size independent of ploidy. We have included a discussion on these issues (lines 481-488).

How does total nuclear size scale with cell size (only myonuclear domain size was shown)?

Response 4b: The sum of all nuclear volumes would depend only on the number of nuclei N since there is, on average, no systematic difference in nuclear volume with cell-size, and the scaling plots would be more or less identical to Fig. 5a, although with a different y -intercept.

5- Initial Comment e.Paragraph starting 349: It is not clear what 'absolute ceiling' means; myonuclear domain sizes in d2w mice could be interpreted as showing nuclei at their maximum capacity, i.e. ceiling? It doesn't make sense to assume that normal muscle fibers under regular conditions would operate at max capacity, but instead optimize for low stress on the system (nice symmetrical gaussian distributions of size parameters). This would allow for the size flexibility that's been observed in so many different muscle studies.

Authors' Response: Here our opinion diverges from the reviewer and we don't understand what the reviewer is referring to by stating: "This would allow for the size flexibility that's been observed in so many different muscle studies.", as no references are provided. As we interpret the current mammalian literature there is rather little flexibility demonstrated. For example, during overload-induced hypertrophy, recruitment of myonuclei precedes growth (see Bruusgaard 2010, PNAS), and

as discussed in the MS hypertrophy is largely prevented when myonuclear accretion is prevented. Although non physiological molecular manipulations (e.g. myostatin blocking) display large fibers with few nuclei, they have impaired function with reduced specific force, reduced myofibrillar apparatus, and they even do not taste good (Belgian blue). The flexibility during developmental growth observed here and in the accompanying paper is discussed in both papers. Please note that our Discussion has been extensively rewritten.

New Comment: The current mammalian literature that was being referred to include the review by Murach et al., 2018 and many references therein, including Kadi et al., 2004, Verney et al., 2008, and Verdijk et al., 2009. Perhaps these should be discussed?

Response 5: Going back to the initial comment by the reviewer on this point, and looking carefully at the relevant paragraph we realize we might have misled the reviewer as the point was to state that since there is no absolute ceiling, this is indeed an example of flexibility of myonuclear domains in the experimental developing muscle. We have rewritten the paragraph for clarity and avoid the term “absolute ceiling” (lines 420-427).

To the more general discussion of the flexibility in myonuclear domains in adults we have included a paragraph and the reference to address the views of the Murach et al (lines 470-474). The review is however related to that group’s hypotheses on a different primary role for satellite cells namely “muscle repair and extracellular matrix remodeling, and not necessarily a precursor to fusion for augmenting transcriptional capacity during adult muscle fiber hypertrophy”, which is rather controversial. We also disagree with several other notions in the review e.g. that nuclear accretion should not pertain to type 2 fibers, on the contrary, most of the evidence for nuclear accretion preceding/accompanying hypertrophy is indeed data from type 2 fibers.

The three original papers mentioned by the reviewer here all focus on satellite cells, not myonuclei, and they don’t use precise methods for assessing the number of myonuclei, for example Kadi and Verney just use hematoxylin on histological sections, and some of the alleged myonuclei are not myonuclei judged from the micrographs as we see it. One of the strengths of our approach is that we have more precise methods for identifying true myonuclei within single fibers. The lack of precise identification of myonuclei has led to some of the controversies in the field, but new tools are now available which might resolve some of the issues.

It is an interesting, but complicated debate, so we feel that a further more technical in depth discussion would be more suited for a review.

**Please check through the text for grammar and clarity
As examples:**

**Line 105 “was ranging” should be ranged
Line 106 “displaying” should be displayed**

Line 372 “The exceptions to similarity in scaling exponents, were...

Line 380-81 “ ..a cytoarchitectural reminiscence of cell size ...”

We apologize for the errors, and have corrected the ones pointed out, and also gone over the MS for other language problems.

Reviewer #2

Initial:

p. 19 1st para It is stated that the larger myonuclear domains in larger muscle cells might be a factor limiting muscle fiber hypertrophy. How does this fit with “double muscle” myostatin knock out mice with large muscle volumes in muscles with very large myonuclear domains?

Authors’ Response: We refer to this (Omairi, et al., 2016; Matsakas, et al., 2012; Metzger, et al., 2012; Bruusgaard, et al., 2005). Myostatin muscles have low specific force, are scarcer in sarcomeres, and even does not taste that good.

New comment: I find the response to my comment regarding myostatin $-/-$ mice surprising. These mice have very large muscle fiber sizes in spite of proportionally low number of myonuclei. This does not fit with the notion that myonuclei number is a general limiter of fiber growth. The muscle fibers of these mice are strong but not in proportion to size resulting in lower specific tensions. However, this study does not include information on regulation of muscle contraction or contractile protein content in any of the other muscles included in this study and is solely focusing on myonuclear number vs. fiber size. The authors comment regarding the taste of myostatin deficient muscles is irrelevant

Response: We agree that the discussion on the issues of a putative “functional penalty” and the myostatin animals was a bit short and flimsy. We have added a new discussion paragraph in order to make our point better (Lines 428-439).

Reviewers' Comments:

Reviewer #1:

Remarks to the Author:

The readability of the manuscript is much improved. The authors make their points in a clearer way.

Nevertheless:

1-The authors should make it clear their data are generated, not from the whole fiber, but from fiber segments. For their analysis, they didn't reconstruct an entire fiber to my knowledge.

As examples: "In Fig. 1 m-r individual fiber data are plotted as a function of fiber size". It is the fiber segment size yes?

Another example: "The number of nuclei per mm was not affected by the denervation (Fig. 3b and f), in agreement with our previous conclusions from wide field microscopy"

Please amend, "The number of nuclei per mm was not affected by the denervation (Fig. 3b and f), in the segments analyzed, in agreement with our previous conclusions from wide field microscopy"

Please correct throughout the document.

2- Segment choice: The authors did not measure NMJ and MTJ segments.

The authors are making an assumption that the NMJ and MTJ nuclei don't contribute much to producing non-synaptic or non-myotendinous junction proteins, thus rationalizing their choice of not including NMJ and MTJ areas of the muscle. Support for their assumptions can be checked and cited with all the recently available single nuclear sequencing data from different groups rather than insitu data sets with a few target genes. The ramifications of the approach ("not including these nuclei") should be discussed in more detail.

3. The authors need to include how they measured nuclear volume using Dapi. The figure just indicates nuclear fluorescence. What did the authors actually measure? There is nothing mentioned in the methods section. This matters.

As example: If they used DAPI intensity as in a 'sum of all DAPI pixel values' that should equal the amount of DNA/nucleus, so in this case any changes in intensity should indicate differences in DNA content per nucleus. If the authors used some kind of 'mean intensity value' then that should equal the density of DNA. In this case (and assuming that DNA content is the same in all the nuclei), lower values would indicate bigger nuclei, while higher values indicate smaller nuclei. Relationships between DNA content and nuclear size are found in papers including 'Mechanisms of Nuclear Size Regulation in Model Systems and Cancer' by Predrag Jevtić and Daniel L. Levy (2014). Please clarify.

Reviewer #2:

Remarks to the Author:

The authors have responded adequately to my critique

Final rebuttal, our answers in red.

Reviewer #1 (Remarks to the Author):

The readability of the manuscript is much improved. The authors make their points in a clearer way.

Nevertheless:

1-The authors should make it clear their data are generated, not from the whole fiber, but from fiber segments. For their analysis, they didn't reconstruct an entire fiber to my knowledge.

As examples: "In Fig. 1 m-r individual fiber data are plotted as a function of fiber size". It is the fiber segment size yes?

Another example: "The number of nuclei per mm was not affected by the denervation (Fig. 3b and f), in agreement with our previous conclusions from wide field microscopy"

Please amend, "The number of nuclei per mm was not affected by the denervation (Fig. 3b and f), in the segments analyzed, in agreement with our previous conclusions from wide field microscopy"

Please correct throughout the document.

We have corrected the specific examples, and in other places throughout the document, and improved the description of the methods.

2- Segment choice: The authors did not measure NMJ and MTJ segments.

The authors are making an assumption that the NMJ and MTJ nuclei don't contribute much to producing non-synaptic or non-myotendinous junction proteins, thus rationalizing their choice of not including NMJ and MTJ areas of the muscle. Support for their assumptions can be checked and cited with all the recently available single nuclear sequencing data from different groups rather than insitu data sets with a few target genes. The ramifications of the approach ("not including these nuclei") should be discussed in more detail.

To accommodate the reviewers concern, we have given more detail in the method chapter (end of the paragraph about Preparation and imaging of developmental cells from EDL), and removed any "assumptions" about the expression profile of specialized nuclei that the reviewer seems to regard as controversial (1 paragraph of results). The reviewer is, not giving any references for sequencing data he/she is referring to, and we are only aware of non-peer reviewed preprints that might possibly have a bearing on what the reviewer is referring to. Our main point, however, is that the specialized nuclei are very few and therefore unlikely to be important for our conclusions in this paper. We feel that the text describing the approach now make it very clear to the reader how our data were acquired, and hence can be evaluated.

3. The authors need to include how they measured nuclear volume using Dapi. The figure just indicates nuclear fluorescence. What did the authors actually measure? There is nothing mentioned in the methods section. This matters.

As example: If they used DAPI intensity as in a 'sum of all DAPI pixel values' that should equal the amount of DNA/nucleus, so in this case any changes in intensity should indicate differences in DNA content per nucleus. If the authors used some kind of 'mean intensity value' then that should equal the density of DNA. In this case (and assuming that DNA content is the same in all the nuclei), lower values would indicate bigger nuclei, while higher values indicate smaller nuclei. Relationships between DNA content and nuclear size are found in papers including 'Mechanisms of Nuclear Size Regulation in Model Systems and Cancer' by Predrag Jevtić and Daniel L. Levy (2014). Please clarify.

We have made more detailed the method description the DNA/nuclear volume measurements, and also discuss caveats as replied in the previous round of reviews, and as already pointed out we respectfully think that this investigation is somewhat peripheral to our main claims about common properties of scaling between the nuclear number and fiber size across species and developmental age. The measurements of nuclear volume were instigated by the reviewer based on the 3D reconstructions he/she pointed out we had in stock, and we clearly highlight that our measurements are a mere proxy of nuclear volume, but that is commonly used in the field. A full account of nuclear quality for example by quantitative fluorescence assessing DNA concentration, as the reviewer seems to ask for, would require different methodology and a separate paper, and as we have pointed out previously this is out of scope for this paper.

Reviwer #2 had no further comments